# Latest Permian carbonate-carbon isotope variability traces heterogeneous organic carbon accumulation and authigenic carbonate formation

Martin Schobben[1,2], Sebastiaan van de Velde[3], Jana Suchocka[2], Lucyna Leda[2], Dieter Korn[2], Ulrich Struck[2], Clemens Vinzenz Ullmann[4], Vachik Hairapetian[5], Abbas Ghaderi[6], Christoph Korte[7], Robert J. Newton[1], Simon W. Poulton[1], and Paul B. Wignall[1]

[1]School of Earth and Environment, University of Leeds, Woodhouse Lane, Leeds, LS2 9JT, United Kingdom
[2]Museum für Naturkunde, Leibniz-Institut für Evolutions- und Biodiversitätsforschung, Invalidenstr. 43, D-10115 Berlin, Germany
[3]Analytical, Environmental and Geochemistry, Vrije Universiteit Brussel, Pleinlaan 2, 1050, Brussels, Belgium
[4]College of Engineering, Mathematics and Physical Sciences, Camborne School of Mines, University of Exeter, Penryn Campus, Penryn, Cornwall TR10 9FE, United Kingdom
[5]Department of Geology, Khorasgan (Esfahan) Branch, Islamic Azad University, P.O. Box 81595-158, Esfahan, Iran
[6]Department of Geology, Faculty of Sciences, Ferdowsi University of Mashhad, Azadi Square, 9177948974, Mashhad, Iran
[7]Department of Geosciences and Natural Resource Management, University of Copenhagen, Øster Voldgade 10, DK-1350 Copenhagen, Denmark

*Correspondence to:* Martin Schobben (m.schobben@leeds.ac.uk or schobbenmartin@gmail.com)

**Abstract.** Bulk-carbonate carbon isotope ratios are a widely applied proxy for investigating the ancient biogeochemical carbon cycle. Temporal carbon isotope trends serve as a prime stratigraphic tool, with the inherent assumption that bulk micritic carbonate rock is a faithful geochemical recorder of the isotopic composition of seawater dissolved inorganic carbon. However, bulk-carbonate rock is also prone to incorporate diagenetic signals. The aim of the present study is to disentangle primary trends from diagenetic signals in carbon isotope records which traverse the Permian–Triassic boundary in marine carbonate-bearing sequences of Iran and South China. By pooling newly produced and published carbon isotope data we confirm that a global first-order trend towards depleted values exists. However, a large amount of scatter is superimposed on this geochemical record. In addition, we observe a temporal trend in the amplitude of this residual $\delta^{13}C$ variability, which is reproducible for the two studied regions. We suggest that (sub)seafloor microbial communities and their control on calcite nucleation as well as on ambient porewater dissolved inorganic carbon-$\delta^{13}C$ pose a viable mechanism to induce bulk-rock $\delta^{13}C$ variability. Numerical model calculations highlight that early diagenetic carbonate rock stabilization, and linked carbon isotope alteration, can be controlled by organic matter supply and subsequent microbial remineralization. A major biotic decline among Late Permian bottom-dwelling organisms facilitated a spatial increase in heterogeneous organic carbon accumulation. Combined with low marine sulfate, this resulted in varying degrees of carbon isotope overprinting. A simulated time series suggests that a 50% increase in the spatial scatter of organic carbon relative to the average, in addition to an imposed increase in the likelihood of sampling cements formed by microbial calcite nucleation to one out of 10 samples, is sufficient to induce the observed signal of carbon isotope variability. These findings put constraints on the application of Permian-Triassic carbon isotope chemostratigraphy

based on whole-rock samples, which appears less refined than classical biozonation dating schemes. On the other hand, this signal of increased carbon isotope variability concurrent with the largest mass extinction of the Phanerozoic may inform about local carbon cycling mediated by spatially heterogeneous (sub)seafloor microbial communities under suppressed bioturbation.

## 1   Introduction

Carbon isotopes in carbonate rock are a pivotal tool for understanding the ancient biogeochemical carbon cycle, as well as a stratigraphic aid in determining the age of sedimentary deposits. Individual fossil carbonate shells are the preferred recorder of the isotope composition of marine dissolved inorganic carbon (DIC). However, some deposits, such as those of Precambrian age, or those which were formed during biotic crises or where shelly fossils are absent, are not suitable for such a single-component approach (Veizer et al., 1999; Prokoph et al., 2008). Under these circumstances, bulk-rock samples are a widely

used alternative recorder (Saltzman, 2001; Brand et al., 2012a, b). This approach has attracted criticism due to the complex multicomponent nature and high potential for diagenetic alteration, which may result in a mixture of primary and secondary diagenetic signals. Major sources of contamination that are known to affect the primary marine carbon isotope signal of bulk-carbonate rock are degradation of organic matter and methane oxidation (Marshall, 1992; Reitner et al., 2005; Wacey et al., 2007; Birgel et al., 2015).

Carbonate sediments are especially prone to chemical alteration during early compaction and lithification, when the highly porous sediment and unstable carbonate polymorphs (e.g., aragonite and high-Mg calcite) are subjected to cementation and dissolution (Bathurst, 1975; Irwin et al., 1977; Marshall, 1992). The potential for diagenetic alteration of biogenic high-Mg calcite and aragonite results from elevated substitution of foreign ions in the lattice and a higher degree of dislocations (Busenberg and Plummer, 1989).

On carbonate platforms subjected to rapid sea level fluctuations, lithification might be controlled by interaction with meteoric fluids and oxidized terrestrial organic matter (Brand and Veizer, 1980, 1981; Banner and Hanson, 1990; Knauth and Kennedy, 2009). The latter mode of carbonate rock cementation with allochthonous dissolved carbonate sources (meteoric water) in a relatively open system has long been viewed as the predominant process of carbonate rock stabilization (Bathurst, 1975). These notions were fuelled by an over-representative focus on Pleistocene carbonate platforms (e.g., the Bahamas) and their

associated diagenetic stabilization under influence of glacial–interglacial induced sealevel fluctuations (James and Choquette, 1983). However, this mechanism is less relevant for rock lithified under greenhouse conditions, when sealevel changes are dampened (Bathurst, 1993; Munnecke and Samtleben, 1996; Melim et al., 2002).

Alternatively, carbonate rock alteration is driven by marine-derived fluids that evolved through interaction with precursor sediments and ambient chemical conditions (Munnecke and Samtleben, 1996; Melim et al., 2002). Foremost, the reactive

and biologically active upper-section of the sediment column can be envisioned to leave an imprint on carbonate chemistry, including the carbon isotope composition. This carbon isotope signal will be retained in post-depositional carbonate cements, and is controlled by organic carbon (OC) fluxes, pH, alkalinity, and the nature of the (sub)seafloor microbial communities (Melim et al., 2002; Reitner et al., 2005; Wacey et al., 2007; Birgel et al., 2015; Schobben et al., 2016; Zhao et al., 2016).

A more nuanced view on the nature of bulk-rock carbon isotope composition would be to envision diagenetic overprinting on carbonate rock chemistry as a spectrum, where pure primary and strictly diagenetic endmember states are considered as a continuum, with bulk-rock chemistry falling somewhere in between these two extremes (cf. Marshall, 1992). In this case, the aim should be to discern the degree to which an original imprint could have been retained in the bulk-rock signal. Embracing both ends of this spectrum could illuminate new aspects of marine sedimentary systems and ancient ocean chemistry, as well as providing a more-refined understanding of the mechanisms that promote the diagenetic alteration of carbonate sediments. Clear examples of the importance of interpreting diagenetic signals have been given by the assertion that authigenic carbonate production might play a primary role in the global biogeochemical carbon cycle (Schrag et al., 2013), and in the isotopic imprint of terrestrial biomass on Precambrian carbonates (Knauth and Kennedy, 2009).

## 2    Background: Permian–Triassic carbonate-carbon isotope records

Temporal trends in carbonate-carbon isotope composition have been recorded at varying time-scales, from million-year secular trends (Veizer et al., 1999; Zachos et al., 2001) to dramatic submillion year events, such as at the Paleocene–Eocene Thermal Maximum (Dickens et al., 1995). The sedimentary record of the Permian–Triassic (P–Tr) boundary interval, marked by the largest mass extinction of the Phanerozoic (Erwin, 1993; Alroy et al., 2008), is also characterised by pronounced carbon isotope excursions, which are almost exclusively recorded in bulk-rock. However, chemical signatures from this time period include records with a wide variety of amplitudes and shapes, and span centimetres to metres of rock sequence (e.g., Xie et al., 2007; Korte et al., 2010). Equally diverse are the proposed drivers of these geochemical signals, ranging from prolonged episodes of volcanism, eustasy-controlled erosion of organic-rich shelf sediments, to abrupt blooms of methane-producing microbes (Renne et al., 1995; Berner, 2002; Rothman et al., 2014).

Variable amplitude carbon isotope excursions in shelf to basin transects have been linked to an intensified biological pump (Meyer et al., 2011; Song et al., 2013). This notion contrasts with the viewpoint of collapsed primary productivity (or a "Strangelove" Ocean) as a driver for the demise of bottom-dwelling and infaunal marine biota (Rampino and Caldeira, 2005). The loss of these geobiological agents resulted in a reduced thickness of the sedimentary mixed layer and a global increase in the occurrence of laminated sediments (Wignall and Twitchett, 1996; Hofmann et al., 2015). Contrary to disrupted primary productivity, these geological features would also agree with a scenario of enhanced OC remineralization and resulting widespread marine de-oxygenation (Meyer et al., 2008, 2011). These contrasting scenarios of global-scale environmental deterioration are difficult to reconcile. Nevertheless, the reduction of sediment mixing is an unequivocal feature that warrants consideration when interpreting P–Tr chemical records.

An elevated oceanic carbonate inventory is a predicted aspect of an ocean without pelagic calcifiers, which only started proliferating during the Mesozoic, and would effectively buffer short-term perturbations to P–Tr marine carbon isotopes (Kump and Arthur, 1999; Zeebe and Westbroek, 2003; Rampino and Caldeira, 2005; Ridgwell, 2005; Payne et al., 2010). This does not, however, exclude local departures from this dynamic equilibrium or depth-related isotope differences, such as those forced by the biological pump (Kump and Arthur, 1999; Rampino and Caldeira, 2005). High carbonate ion concentrations are  invoked

to explain the advent of (microbial) carbonate seafloor structures (e.g., thrombolites, stromatolites and fan-shaped structures) in the extinction aftermath and the ensuing Early Triassic recovery phase (Baud et al., 1997; Rampino and Caldeira, 2005; Riding and Liang, 2005; Pruss et al., 2006; Kershaw et al., 2007, 2009; Leda et al., 2014). While conditions favoured the formation of these structures, poorly buffered calcified metazoans (e.g., brachiopods and corals) were proportionally more affected by the end-Permian mass extinction (Knoll et al., 2007). This biotic shift has been interpreted as a prevalent carbonate factory turnover, from skeletal to microbial (e.g., Kershaw et al., 2009).

A carbonate factory turnover, reduced sediment mixing and increased OC sinking fluxes invite consideration of a scenario where these individual parameters act as synergistic effects on carbonate formation and diagenetic stabilization. Moreover, the imprint of $^{12}$C-depleted carbon on latest Permian carbonate rock by the introduction of authigenic carbonate has been connected to a systematic carbon isotope offset between bulk-rock and brachiopod shells (Schobben et al., 2016). Diagenetic and authigenic carbonate sources can, however, result in a range of carbon isotope values, relating to the specific microbial community and sedimentary environment (Irwin et al., 1977). Hence, we hypothesize that spatial carbon isotope variability at the P–Tr boundary interval relates to an increased importance of microbially controlled calcite nucleation. Moreover, we postulate that organic carbon sinking fluxes and subsequent *in situ* remineralization by microbes determine the trajectory of carbonate rock stabilization. Combined geographic-differential OC sinking fluxes (km-scale) and sediment mixing (cm-scale) might have generated spatially heterogeneous dispersion of organic carbon. This spatial pattern of OC distribution links to observed carbonate carbon isotope variability by modulating the chance of sampling variable isotope signals along lateral lithological transects. We adopt this conceptual model as a working hypothesis for interpreting stratigraphic carbon isotope patterns in P–Tr carbonate rock. In addition, however, this scenario likely has implications for the interpretation of bulk-rock carbon isotope patterns during other periods of the Precambrian and Phanerozoic.

To carry out this investigation, we studied carbonate-bearing sequences of Permian to Triassic strata located in Iran and China using a compilation of published and new data. The effect of microbial metabolism on sediments and porewater can be numerically approximated by reactive-transport models (Soetaert et al., 1996; Boudreau, 1997; Meysman et al., 2003; van de Velde and Meysman, 2016). Similarly, diagenetic models have proven useful in delineating trajectories of bulk-rock Sr and Ca isotope stabilization (Fantle and DePaolo, 2006, 2007). For the purpose of our study, we will combine aspects of these models in an effort to estimate the potential for microbially mediated spatial modification of bulk-rock carbon isotope signals.

## 3 Materials and methods

### 3.1 Bulk-carbonate carbon isotope records

#### 3.1.1 Material selection and stable carbon isotope measurements

The P–Tr limestone sequences of Iran are particularly suited to study spatial $\delta^{13}C_{carb}$ variations, as lateral lithological homogeneity excludes a strong control from palaeoceanographic conditions or selective preservation potential (Supplementary Information). Besides the classical P–Tr sites of, e.g., Shahreza, Zal and Ali Bashi of Iran, we sampled several other sites,

for which we present the first $\delta^{13}C_{carb}$ results. These sites include the Aras Valley profile (39.015°N, 45.434°E) about 19 km WNW of the towns of Dzhulfa (Azerbaijan) and Julfa (Iran), four parallel sections in the Baghuk mountains (section A: 31.563°N, 52.438°E; section 1: 31.567°N, 52.444°E; section C: 31.567°N, 52.443°E; section J: 31.565°N, 52.441°E) 50 km NNW of Abadeh and 140 km SSE of Esfahan, as well as the Asadabad succession (31.848°N, 52.181°E). Limestone, marl-
stone and calcareous shale (>20 % $CaCO_3$) beds were collected bed-by-bed. Powder aliquots were produced by microdrilling fresh rock surfaces to avoid sampling of obvious late diagenetic calcite veins and weathered surfaces.

A total of 463 stable carbon isotope measurements were made at the University of Copenhagen (UC) and the Museum für Naturkunde (MfN), Berlin. Glass reaction vessels (LABCO®) containing the sample powders were flushed with helium, and carbonate was left to react with 50 (UC) or 30 (MfN) µl of anhydrous phosphoric acid (~102 %), for at least 1.5 h. Carbon and
oxygen isotope values were measured from resulting $CO_2$ using an IsoPrime triple collector Isotope Ratio Mass Spectrometer in continuous flow setup (UC) or a Thermo Finnigan Gasbench (GS) II linked to a THERMO/Finnigan MAT V isotope ratio mass spectrometer (IRMS) (MfN). Pure $CO_2$ (99.995 %), calibrated against international IAEA standards (NSB-18 and NSB-19), was used as a reference gas. At the UC, results were corrected for weight-dependent isotope ratio bias using multiple measurements of an in-house standard LEO (Carrara Marble, $\delta^{13}C$ = 1.96‰) covering the entire range of signal intensities
encountered in the samples. External reproducibility was monitored by replicate analysis of the in-house standards LEO and Pfeil STD (Solnhofen Limestone), at the UC and MfN respectively. Long-term accuracy was better than 0.1‰ ($2\,SD$) (and better than 0.2‰ $2\,SD$ for oxygen isotopes). All carbon isotope values are reported in ‰ relative to VPDB, and in standard $\delta$-notation.

### 3.1.2   Complementary data collection

Our new data are complemented by published bulk-carbonate $\delta^{13}C$ data from multiple P–Tr localities in Central Iran (Abadeh (Heydari et al., 2000; Korte et al., 2004b; Horacek et al., 2007; Korte et al., 2010; Liu et al., 2013) and Shahreza (Korte et al., 2004a; Heydari et al., 2008; Richoz et al., 2010), as well as Northwest Iran (Ali Bashi mountains and Zal; Baud et al., 1989; Korte et al., 2004c; Korte and Kozur, 2005; Kakuwa and Matsumoto, 2006; Horacek et al., 2007; Richoz et al., 2010; Schobben et al., 2016). For a more global perspective and comparative approach, we additionally extracted published data from the P–Tr
Global Stratigraphic Section and Point (GSSP) at Meishan, China (Baud et al., 1989; Chen et al., 1991; Xu and Yan, 1993; Hoffman et al., 1998; Jin et al., 2000; Cao et al., 2002; Gruszczyński et al., 2003; Zuo et al., 2006; Kaiho et al., 2006; Huang et al., 2007; Riccardi et al., 2007; Xie et al., 2007). Data was extracted from tables and supplementary files or provided by the authors, and where necessary read from figures with the open-source software *xyscan* (Ullrich, 2016). This compilation adds 2077 data points to our new dataset. The analytical uncertainty of the collected data, when given, ranges between 0.02 and
0.2‰. We included replicate studies on the same site, as this data is potentially a valuable asset to test the reproducibility of stable isotope investigations and to shed light on the effect of the bulk-rock multicomponent nature on $\delta^{13}C_{carb}$ composition.

**Table 1.** Correlative biostratigraphic units

| Unit | Iran | China[*] |
|------|------|----------|
| K | *Neospathodus dieneri* | *Stepanovites kummeli* |
| J | *Hindeodus postparvus* | *Clarkina tulongensis-Clarkina planata* |
| I | *Isarcicella isarcica* | *Isarcicella isarcica* |
|   | *Isarcicella staeschei* | *Isarcicella staeschei* |
| H | *Hindeodus lobata* | |
|   | *Hindeodus parvus* | *Hindeodus parvus* |
| G | *Merrillina ultima - Stepanovites mostleri* | |
|   | *Hindeodus praeparvus - Hindeodus changxingensis* | *Hindeodus changxingensis* |
| F | *Clarkina hauschkei* | |
|   | *Clarkina abadehensis* | *Clarkina meishanensis* |
| E | *Clarkina yini* | *Clarkina yini* |
|   | *Clarkina nodosa* | |
|   | *Clarkina bachmanni* | |
| D | *Clarkina changxingensis* | *Clarkina changxingensis* |
| C | *Clarkina subcarinata* | *Clarkina wangi - Clarkina subcarinata* |
| B | *Clarkina orientalis* | *Clarkina orientalis* |
| A | *Clarkina trancaucasica* | |
|   | *Clarkina leveni* | |

[*] Meishan P–Tr GSSP. See the Supplementary Information file for references regarding the biostratigraphy studies considered to construct this biozonation scheme.

### 3.1.3 Data projection

To facilitate a direct comparison of $\delta^{13}C_{carb}$ between different sites with differences in total sediment thickness, we converted the stratigraphic height to a dimensionless timeline. This approach preserves a more direct connection to the sampled rock sequence, rather than converting to absolute ages or maintaining original stratigraphic heights. The conversion to a dimension-
less time grid was carried out by using the lower and upper bounds of individual conodont assemblage biozones as tie-points to which a relative distance for an individual $\delta^{13}C_{carb}$ datapoint is assigned (Supplementary Information). The chronological scheme used here divides the P–Tr interval into biostratigraphic units (A–K), which enables correlation of individual sequences of both geographic regions (Table 1).

To evaluate trends in the collected data, a sliding window with a bandwidth equivalent to the dimensionless grid unit has
been applied to the $\delta^{13}C_{carb}$ data. This method ensures the extraction of temporal trends at equivalent time resolution to the biozonation units (Fig. 1).

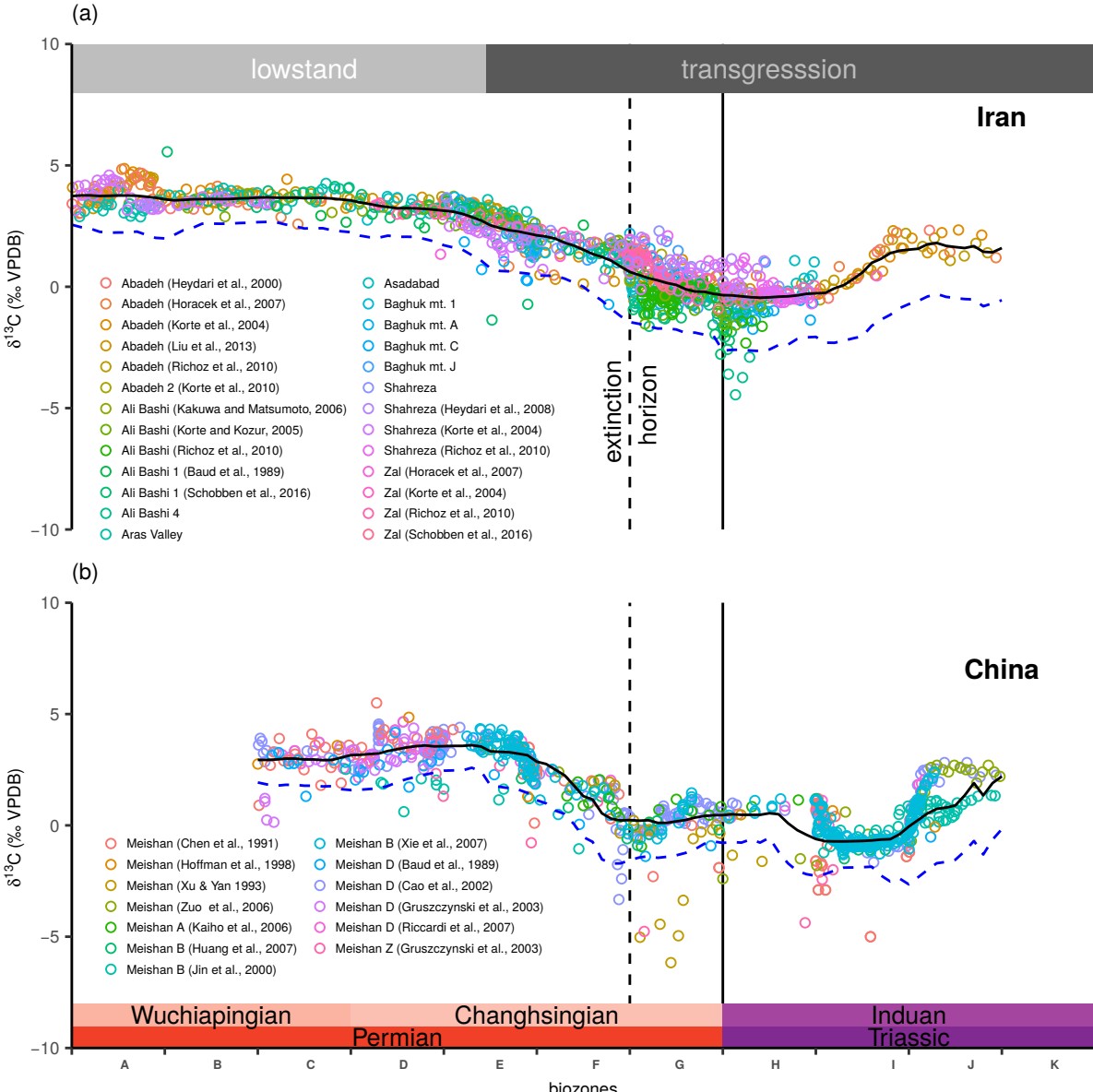

**Figure 1.** Compiled published and new data for multiple P–Tr rock sequences in Iran (a) and China (b). The individual carbonate carbon isotope values are placed on a dimensionless timeline that marries both geographic areas in the most acceptable biochronological scheme (Table 1). The solid black line represents the subsampled median trend line. The dashed blue line depicts the seawater $\delta^{13}C_{DIC}$ curve, as obtained from the time series simulation (Sect. 5). The stratigraphic placement of the sea level changes and the extinction horizon as well as the biostratigraphic framework can be found in the Supplementary Information.

To cancel out the potential for an uneven spread of data points we also applied a subsampling routine on the Iranian and Chinese datasets. Subsampling was performed with the sliding window procedure, as described above, before applying summary statistics. Subsequently, the median values were calculated for each of subsampled $\delta^{13}C_{carb}$ population. For simplification, the resulting median trend line and 95 % confidence intervals (CI) are calculated and visually weighted (Fig. 2) (cf. "Visually Weighted Plot" or "Watercolor Regression"; Hsiang, 2012; Schönbrodt, 2012). The latter is accomplished by calculating a kernel density, and both the drawn median trend line and CI interval are weighted by colour saturation, based on this kernel density value. In addition, the median trend line and the CI interval have been assigned contrasting colours to further enforce the visual signal. Hence, saturated and contrasting parts of the depicted regression curve (i.e., sections with high visual weight) depict areas with the highest fidelity of the temporal geochemical pattern, based on the subsampling routine. In addition to the median trend lines, the same subsampling approach and visual weighting has been applied to calculate and plot the interquartile range (IQR; the value-range containing 50 % of the sample-population) and 95 %-interpercentile range (IPR; the value-range containing 95 % of the sample-population). Graphs are plotted with the open-source programming platform R (R Core Team, 2016) and with the aid of the R packages; *ggplot2* (Wickham, 2009), *reshape2* (Wickham, 2007), *plyr* (Wickham, 2011) and *gridExtra* (Auguie, 2016).

## 3.2 Reactive-transport modelling

### 3.2.1 Geochemical model formulation and biogeochemical reactions

Organic matter availability fuels *in situ* metabolic pathways linked to calcite nucleation. Upon entering the sediment, organic matter is microbially mineralized, which is coupled to the reduction of electron acceptors. These electron acceptors are consumed in a well-defined sequence, based on their thermodynamic energy yield; $O_2$, $SO_4^{2-}$ and methanogenesis (Berner, 1964; Froelich et al., 1979). Each of these microbial metabolisms will imprint a specific carbon isotope signature on the porewater DIC and thus create a potential source for diagenetic alteration of carbonate-C isotope signatures (Irwin et al., 1977).

These mineralization processes can be described by mass balance equations (Eqs 1 and 2), which can subsequently be solved numerically via the *method-of-lines* (Boudreau, 1996; Soetaert and Meysman, 2012). The numerical solutions of these equations solve the age–depth ($t$ and $z$) relationship of deposited sediments in terms of solids ($S_i$), as well as solutes ($C_i$) in pore fluids, and their respective reactions between the solid and liquid phase. At the top of the sediment pile the porewaters communicate with ocean water so that dissolved elements can diffuse according to their concentration gradients. Besides diffusion transport processes, continued sedimentation supplies the sediment with organic carbon and calcium carbonate.

$$\varphi\frac{\partial C_i}{\partial t} = \frac{\partial}{\partial z}(\varphi D_i \frac{\partial C_i}{\partial z} - \varphi\nu C_i) + \sum_k \nu_{i,k}R_k + \varphi\alpha(z)(C_{ow} - C(z)) \tag{1}$$

$$(1-\varphi)\frac{\partial S_i}{\partial t} = \frac{\partial}{\partial z}((1-\varphi)D_b(z)\frac{\partial S_i}{\partial z} - (1-\varphi)wS_i) + \sum_k \nu_{i,k}R_k \tag{2}$$

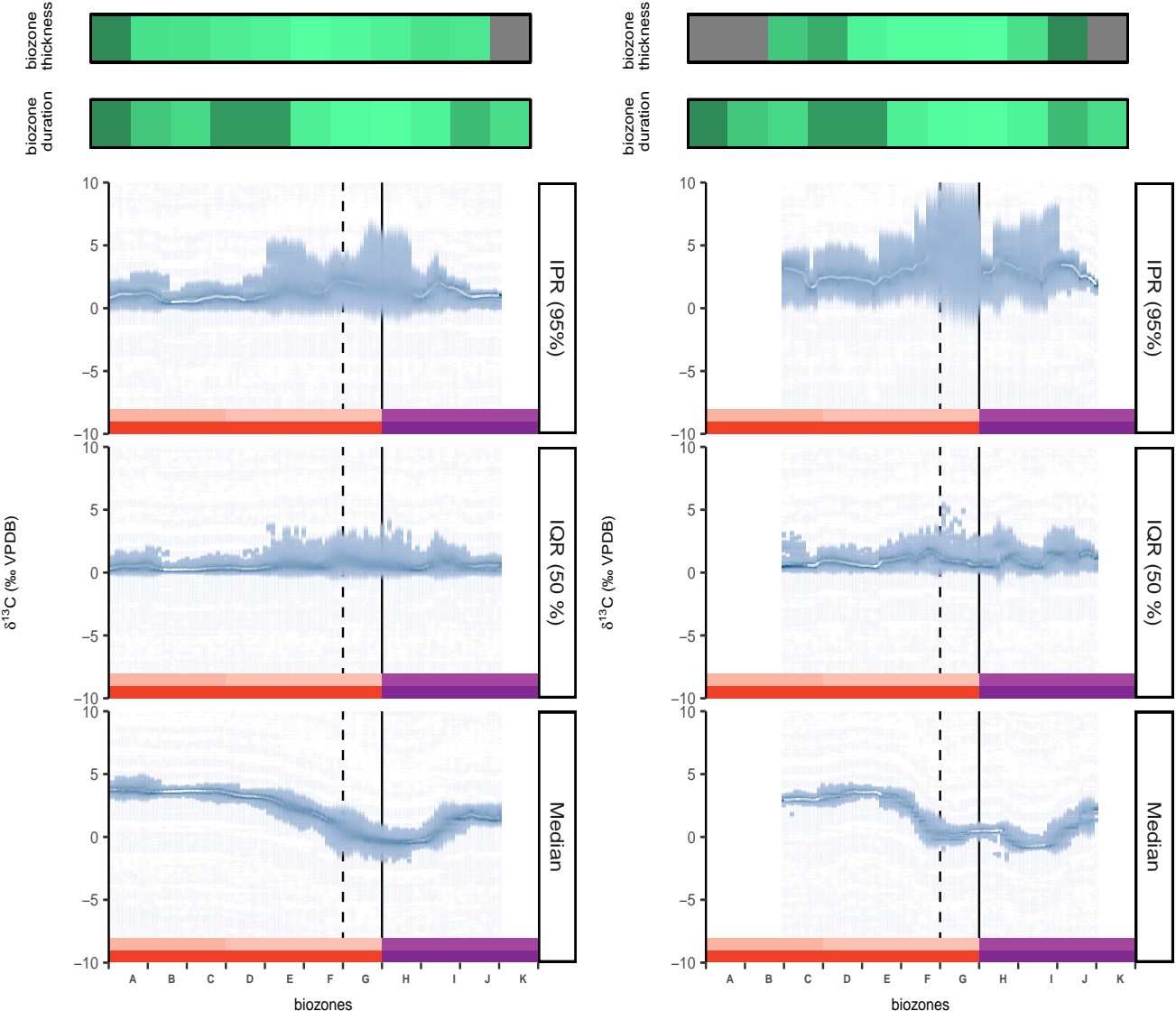

**Figure 2.** Visually weighted data plot for the Iranian (left) and South China (right) P–Tr (sub)localities (Sect. 3.1.3), depicting the median trend line, the IQR (50 %) and the IPR (95 %) $\delta^{13}$C value-ranges. Trends with lowest fidelity are marked by a blurring of colours and less contrasted colours (based on the CI of multiple subsample routines). The dashed and the solid line represent the extinction horizon and P–Tr boundary, respectively. The saturation-level of the green tiles in the upper two panels equals a more extended biozone thickness (0.12–32.00 m) and longer duration (0.02–1.00 My). Grey tiles represent intervals with no available biostratigraphic data. See Fig. 1 for colour-legend of the stratigraphic units.

In equations 1 and 2, $\varphi$ is porosity, $D_i$ the effective diffusion coefficient and $D_b$ and $\alpha(z)$ parameters associated with bioturbation (see below). $\nu_{i,k}$ the stoichiometric coefficient of species $C_i$ in reaction $R_k$. Note that we express reactions as

mol per unit time per volume sediment and concentrations as mol per volume pore water or volume solid phase. Therefore $\varphi$ and $1$-$\varphi$ are introduced as unit conversions. The model includes three different modes of transport; sedimentation (represented by downward advection of solutes $v$ and solids $w$) , molecular diffusion and biological transport (bioturbation). Since we only consider cohesive sediments, the only advective transport is burial, i.e., new sediment is added on top of the sediment column, and sediment at the bottom of the column is buried (transported out of the column). Molecular diffusion for porewater solutes is expressed by Fick's first law, where $D_i$ (the effective diffusion coefficient) is calculated following the definition as given in Boudreau and Meysman (2006).

$$D_i = \frac{D_0}{(1 - 2\ln\varphi)} \tag{3}$$

Where $D_0$ is a function of temperature and salinity and has been calculated with the R package *marelac* (Soetaert et al., 2016) . Bioturbation is implemented as two separate processes; bio-mixing and bio-irrigation (Kristensen et al., 2012), following conventional descriptions (Boudreau, 1984; Meysman et al., 2010). The bio-mixing is modulated over the depth interval to account for the effects of sediment reworking by metazoans in the top layer of the sediment pile. In this formulation it is assumed that $D_b$ remains constant in a layer with thickness $z_b$, after which it attenuates with depth, following an exponential relation with coefficient $\lambda_{D_b}$ and background bio-diffusivity ($D_{b0}$) at the seawater–sediment interface (Soetaert et al., 1996).

$$D_b(z) = D_{b0} \qquad z \leq z_b$$

$$\text{for}$$

$$D_b(z) = D_{b0} \exp\left(\frac{-(z - z_b)}{\lambda_{D_b}}\right) \qquad z > z_b \tag{4}$$

Bio-irrigation exchanges porewater with the overlying water via burrow flushing. This is implemented via a non-local exchange process (Boudreau, 1984).

$$I_{irr}(z) = \alpha(z)(C_{ow} - C(z)) \tag{5}$$

The quantity $\alpha(z)$ represents the depth-dependent irrigation intensity, and the solute concentrations of the bottom water and at depth are given by; $C_{ow}$ and $C(z)$, respectively. The attenuation of bio-irrigation is expressed by an exponential relation.

$$\alpha(z) = \alpha_0 \exp\left(-\frac{z}{x_{irr}}\right) \tag{6}$$

In this formulation, $\alpha_0$ is the irrigation coefficient at the sediment water interface, and $x_{irr}$ is the attenuation depth coefficient. Most of the macrofaunal activity takes place in the upper few centimetres of the sediment (as animals are dependent on food resources that rain down via the water column). Therefore, both bio-mixing and bio-irrigation processes are most intense near the sediment–water interface.

The aim of this model is to give a parsimonious description of the potential effect of early diagenetic reactions on the isotope signature of carbonates. Therefore, we do not consider nitrification or metal cycling, as this would increase the complexity of the model and calculations, and, in addition, this would not fundamentally alter our conclusions. Organic matter can be mineralized via different pathways; aerobic respiration, sulfate reduction and organoclastic methanogenesis (where organic matter acts both as electron donor and acceptor)(Table 2). The reduction of sulfate produces sulfide which can be oxidized by oxygen via canonical sulfide oxidation (CSO) (Meysman et al., 2003; van de Velde and Meysman, 2016). Methanogenesis produces methane, which can be oxidixed by sulphate in a process called anaerobic oxidation of methane (AOM), or by oxygen (ArOM)(Boetius et al., 2000). The mineralization reactions are expressed via standard limitation–inhibition formulations (Soetaert et al., 1996). The Monod constants ($K_{O_2}$ and $K_{SO_4^{2-}}$) in these formulations determine the inhibition or limitation of a certain electron acceptor. For example, at high concentrations of oxygen, oxygen reduction will be efficient and other mineralisation pathways will be inhibited. On the other hand, at lower concentrations, oxygen reduction will be less efficient, and other pathways will gain importance (Soetaert et al., 1996; Van Cappellen and Wang, 1996; Berg et al., 2003; Meysman et al., 2003). This sequential alternation of limitation and subsequent inhibition of metabolic pathways results in chemical zonation of the sediment profile (Froelich et al., 1979). The re-oxidation reactions are given as second-order rate laws where the kinetic constants ($k_{ArOM}$, $k_{AOM}$ and $k_{CSO}$) determine the reaction rate of the respective reactions (Table 2; Boudreau, 1997; Meysman et al., 2003).

To mimic calcium carbonate recrystallization, following an age depth relationship, dissolution and carbonate production reaction rates, expressed as Eq. 7, have been incorporated in the model with no net carbonate dissolution or precipitation (Fantle and DePaolo, 2006, 2007).

$$R_{carb}(t) = \frac{0.4}{f_{dia}} \exp\left(-\frac{t}{0.876 f_{dia}}\right) \tag{7}$$

In contrast to the models of Fantle and DePaolo (2006, 2007) we envision that only a fraction of the whole carbonate rock determines the reactivity towards dissolution and recrystallization. This fraction of 0.2 is considered to be the diagenetic carbonate fraction ($f_{dia}$) of the total rock volume (e.g., Schobben et al., 2016). Specific metabolically produced DIC-$\delta^{13}$C values are assigned for organoclastic microbial sulfate reduction, AOM and organoclastic methanogenesis, see Table 2. These metabolic pathways have been considered to induce calcite nucleation by either changing the surrounding carbonate chemistry, formation of an organic matrix or destroying calcification-inhibiting organic mucus (Berner et al., 1970; Visscher et al., 2000; Arp et al., 2001; Ries et al., 2008; Heindel et al., 2013; Birgel et al., 2015). Peak activity of these metabolic pathways coincides with the most reactive upper layers of the sediment column, i.e., highly porous and high metastable calcite polymorph content (Irwin et al., 1977; Marshall, 1992), and this is mathematically expressed by Eq. 7.

Solute diffusion tracts and sedimentation of the carbon isotopes of DIC and solid-phase carbonate are defined as: $^{13}$C and $^{12}$C, respectively, and are calculated separately. Metabolically introduced carbon ($v_{metabolism}$) by reaction ($R_k$) and its

**Table 2.** Biogeochemical reaction equations and kinetic rate expressions

| Biogeochemical reactions | formulation | $\delta^{13}\mathbf{C_{DIC}}$ | kinetic rate expression |
|---|---|---|---|
| *Primary redox reactions* | | | |
| Oxic respiration | $CH_2O + O_2 \rightarrow HCO_3^- + H^+$ | −25 ‰ | $\frac{[O_2]}{[O_2]+K_{O_2}} k_{min}[CH_2O]$ |
| Organoclastic sulfate reduction | $CH_2O + \frac{1}{2}SO_4^{2-} \rightarrow HCO_3^- + \frac{1}{2}HS^- + \frac{1}{2}H^+$ | −25 ‰ | $\frac{[SO_4^{2-}]}{[SO_4^{2-}]+K_{SO_4^{2-}}} \frac{K_{O_2}}{[O_2]+K_{O_2}} k_{min}[CH_2O]$ |
| Organoclastic methanogenesis | $CH_2O \rightarrow \frac{1}{2}CH_4 + \frac{1}{2}CO_2$ | +15 ‰ | $\frac{K_{SO_4^{2-}}}{[SO_4^{2-}]+K_{SO_4^{2-}}} \frac{K_{O_2}}{[O_2]+K_{O_2}} k_{min}[CH_2O]$ |
| *Secondary redox reactions* | | | |
| Aerobic oxidation of methane | $\frac{1}{2}CH_4 + O_2 \rightarrow \frac{1}{2}CO_2 + H_2O$ | −45 ‰ | $\varphi k_{ArOM}[O_2][CH_4]$ |
| Anaerobic oxidation of methane | $CH_4 + SO_4^{2-} \rightarrow HCO_3^- + HS^- + H_2O$ | −45 ‰ | $\varphi k_{AOM}[SO_4^{2-}][CH_4]$ |
| Canonical sulfur oxidation | $HS^- + 2O_2 \rightarrow SO_4^{2-} + H^+$ | – | $\varphi k_{CSO}[O_2][HS^-]$ |
| *Carbonate precipitation/dissolution* | | | |
| Production diagenetic calcite | $Ca^{2+} + 2HCO_3^- \rightarrow CaCO_3 + CO_2 + H_2O$ | $\epsilon_{carb-DIC}$ | $\frac{K_{O_2}}{[O_2]+K_{O_2}} R_{carb}$ |
| Dissolution calcite | $CaCO_3 + CO_2 + H_2O \rightarrow Ca^{2+} + 2HCO_3^-$ | $\epsilon_{carb-DIC}$ | $\frac{K_{O_2}}{[O_2]+K_{O_2}} R_{carb}$ |

Metabolism-produced carbon isotope values according to Irwin et al. (1977), except organoclastic methanogenesis-sourced $CH_4$ subsequently employed by AOM and ArOM which is a conservative estimate between −65 ‰ and near-quantitative sedimentary OC conversion to $CH_4$, thereby approaching the parent OC-C isotope composition of −25 ‰.

respective effect on porewater DIC-carbon isotope composition is formulated by Eqs 8 and 9 for the heavy and light isotope, respectively.

$$v_{metabolism} = R_k \frac{1}{1 + r_k} \tag{8}$$

$$v_{metabolism} = R_k \frac{r_k}{1 + r_k} \tag{9}$$

We assign a $^{13}C/^{12}C$-ratio ($r$) to the metabolism-specific produced DIC from the organic carbon remineralisation reactions according to the data in Table 2. The mathematical formulation of metabolism-introduced DIC ensures that the isotope composition of ambient porewater DIC is altered, whereas mass-balance in the model carbon in- and output is maintained.

### 3.2.2 Numerical solutions and parameters

The model includes nine state variables: organic matter ($CH_2O$), solid carbonate ($^{13}C_{carb}$ and $^{12}C_{carb}$), DIC ($^{13}C_{DIC}$ and $^{12}C_{DIC}$), dissolved oxygen ($O_2$), dissolved sulfate ($SO_4^{2-}$), dissolved methane ($CH_4$), and dissolved sulfide ($HS^-$). We numerically solved Eqs 1 and 2 on the open-source programming platform R (R Core Team, 2016) with a finite difference approach by expanding the spatial derivatives of partial differential equations over a sediment grid (Boudreau, 1997). By application of

the R package *ReacTran* (Soetaert and Meysman, 2012), a sediment grid of 10 m thick with 20,000 layers of identical thicknesses was generated. The finite difference approach transforms the partial differentials into ordinary differentials which are subsequently integrated by a stiff solver routine *vode* (Brown and Hindmarsh, 1989), within the R package *DeSolve* (Soetaert et al., 2010). Boundary conditions were chosen to reflect Permian oceanic conditions based on the current state of research (Sect. 2), and complemented with current knowledge of shallow marine environments. The upper boundary conditions were set to concentrations for normal modern bottom water conditions; $0.28 \, \mu\mathrm{mol} \, \mathrm{cm}^{-3}$ $O_2$ and zero for the reduced species ($CH_4$ and $HS^-$). A lower seawater $SO_4^{2-}$ value of $4 \, \mu\mathrm{mol} \, \mathrm{cm}^{-3}$ (compared to the modern value of $28 \, \mu\mathrm{mol} \, \mathrm{cm}^{-3}$) was chosen, following evidence of a reduced global marine S-reservoir in the latest Permian to Early Triassic (Luo et al., 2010; Song et al., 2014; Schobben et al., 2017). On the other hand, DIC was set at $4.5 \, \mu\mathrm{mol} \, \mathrm{cm}^{-3}$ (higher then the modern value of $2.2 \, \mu\mathrm{mol} \, \mathrm{cm}^{-3}$), based on model calculations of a Late Permian ocean without pelagic calcifiers (Ridgwell, 2005). The solid phases entering the model at the top of the sediment stack are set $730 \, \mu\mathrm{mol} \, \mathrm{cm}^{-2} \, \mathrm{y}^{-1}$ for both the OC flux ($F_{OC}$) and the carbonate flux ($F_{carb}$) in the baseline model, which are typical average shelf sedimentation values (Müller and Suess, 1979; Reimers and Suess, 1983; Sarmiento and Gruber, 2004). For the baseline-condition, sedimentation rates ($v$ and $w$) are fixed at $0.2 \, \mathrm{cm} \, \mathrm{y}^{-1}$ and the $\delta^{13}C$ composition is set at $+5\,\permil$ (VPDB), based on primary carbon isotope values from pristine preserved brachiopod calcite from the Ali Bashi and Meishan sections, as well as sites in northern Italy (Korte et al., 2005; Brand et al., 2012b; Schobben et al., 2014). For the bio-mixing and bio-irrigation parameters, values for the Palaeozoic, proposed by Dale et al. (2016) where chosen; $D_{b0} = 5 \, \mathrm{cm}^{2-} \, \mathrm{y}^{-1}$, $z_b = 2 \, \mathrm{cm}$, $\alpha_0 = 50 \, \mathrm{y}^{-1}$, $x_{irr} = 1 \, \mathrm{cm}$. This will approximate sedimentary conditions of a seafloor inhabited by a Permian benthic community. The relation between the isotope composition of DIC and solid phase carbonate is given by the following thermodynamic relation (Eq. 10), established by Emrich et al. (1970), where we assume temperature ($T$) to be constant at 25 °C

$$\epsilon_{carb-DIC} = 1.85 + 0.035[T(°C) - 20] \tag{10}$$

The lower boundary conditions for both solid and solute species were set at a "no gradient" boundary, ensuring that materials are only transported by burial to deeper parts of the sediment column. Kinetic parameters are summarized in Table 3, and are based on published models.

### 3.2.3 Bulk-carbonate carbon isotope signal imprinting and its sensitivity towards environmental parameters

The previously defined parameters provide a baseline model to test the sensitivity towards certain boundary conditions, in bringing forward carbon isotope compositional changes in the porewaters. Two pathways of carbonate rock formation and stabilization are considered to be of importance in capturing these porewater carbon isotope compositional changes.

1) Diagenetic carbonate alteration by ongoing recrystallization and dissolution with depth after burial, formulated as Eq. 7, and with both processes operating in equilibrium (Fantle and DePaolo, 2006, 2007). Under this mechanism carbonate is constantly exchanged between the solid (carbonate rock) and aqueous phase (porewater DIC) with ongoing burial, thereby to some degree buffering the isotopic perturbations caused by microbial mineralization. This pathway of carbonate sediment

**Table 3.** Parameter values for the kinetic constants of the reactive-transport model

| Constant | Symbol | Unit | Value | Reference |
|---|---|---|---|---|
| *Organic matter reduction* | | | | |
| Decay constant for organic matter | $k_{min}$ | $\text{y}^{-1}$ | 0.1 | a |
| Monod constant oxygen consumption | $K_{O_2}$ | $\mu\text{mol}\,\text{cm}^{-3}$ | 0.001 | b |
| Monod constant sulfate reduction | $K_{SO_4^{2-}}$ | $\mu\text{mol}\,\text{cm}^{-3}$ | 0.9 | b |
| *Oxidation reactions* | | | | |
| Kinetic constant ArOM | $k_{ArOM}$ | $\mu\text{mol}\,\text{cm}^{-3}\,\text{y}^{-1}$ | $10^4$ | c |
| Kinetic constant AOM | $k_{AOM}$ | $\mu\text{mol}\,\text{cm}^{-3}\,\text{y}^{-1}$ | $10^4$ | c |
| Kinetic constant canonical sulfur oxidation | $k_{CSO}$ | $\mu\text{mol}\,\text{cm}^{-3}\,\text{y}^{-1}$ | $10^4$ | b |

References; a, (Fossing et al., 2004); b, (Meysman et al., 2015); c, (Contreras et al., 2013)

progression can be viewed as the classical interpretation of limestone stabilization: dissolution of less stable components and subsequent occlusion of the produced porespaces by cements (Bathurst, 1993; Munnecke and Samtleben, 1996; Melim et al., 2002).

2) Authigenic carbonate addition by near-instantaneous precipitation of carbonate crusts at (or close to) the seafloor, by microbial mat communities. The latter mechanism would simulate the seafloor crust formation commonly observed for the P–Tr transition, which is recognized as a time of unusual carbonate sedimentation; e.g., stromatolites, thrombolites and microbially induced cementation (Baud et al., 1997; Woods et al., 1999; Leda et al., 2014). However, the mechanism directly responsible for this type of biological-induced calcification is ambiguous, and might be related to extracellular polymeric substances and the orientation of functional groups on this organic matrix (creating nucleation sites), destruction of organic compounds that inhibit calcification or a direct consequence of metabolism and associated changes in ambient carbonate chemistry (Visscher et al., 2000; Arp et al., 2001; Bundeleva et al., 2014). We furthermore assume that cryptic forms of carbonate precipitates, e.g., rock-binding cements (e.g., Richoz et al., 2010) might have escaped detection when selecting carbonate rock for carbon isotope studies.

These two pathways are henceforward referred to as diagenetic carbonate alteration and authigenic carbonate addition, respectively. Note, that porewater chemistry and its control on carbonate formation is not explicitly formulated in the model, and is instead expressed by a reaction rate that can vary with age (Eq. 7) (cf. Fantle and DePaolo, 2006, 2007). This formulation is justifiable due the uncertainties in the ultimate control of biological activity in steering calcification, as cited above.

Diagenetic carbonate alteration is determined by taking the evolved solid-phase $\delta^{13}$C at 10 m sediment depth, where carbon isotope exchange between porewater and solid carbonate reaches equilibrium (see also Fantle and DePaolo, 2006, 2007, for a comparative $\text{Sr}$ and $\text{Ca}$ isotope equilibrium with depth). Although transient seawater chemistry changes can occur over the time needed to complete carbonate stabilization (i.e., before reaching equilibrium at 10 m depth), we consider the adopted isotope value to be a good approximation of a partially recrystallized carbonate rock, as diffusion and the majority of metabolic

processes are confined to only a small interval of the sediment column (upper ~1 m of the sediment pile). Authigenic carbonate addition is regarded to take place in the upper 0.1 m of the sediment column by precipitation from porewater DIC that is most severely perturbed by metabolic acitivity, and largerly unbuffered by exchange with carbonate sediments through dissolution and recrystallization. The evolved bulk-rock carbon isotope value is then calculated by mass balance of primary and authigenic carbonate ($f_{auth}$) components.

An important parameter inducing changes in the vertical structure of a diagenetic redox-profile, and ultimately carbonate-carbon isotope composition during carbonate rock formation and stabilization, is the $F_{OC}$. By systematically changing this parameter we test the sensitivity of the diagenetic baseline model to the OC sinking flux and the consequential organic matter remineralization trajectories, i.e., the dominant microbial communities involved in local *in situ* C-cycling. The effect of the OC sedimentary regime on the carbonate isotope system is probed by looking at the C isotope offset the primary carbonate precipitate (settling through the water column) and the bulk-rock endmember ($\Delta^{13}\mathrm{C}_{\mathrm{primary-bulk}}$). Besides the OC flux, we performed sensitivity experiments for sedimentation rate ($w$ and $v$), the concentrations of bottom water DIC and sulfate as well as the fraction of diagenetic/authigenic carbonate ($f_{dia}/f_{auth}$) incorporated in the bulk-rock. For each separate sensitivity experiment we return the changing parameter to the initial value, as defined for the baseline model (Sect. 3.2.2), except for the parameter of interest and the OC flux. In addition, we tested the influence of bio-diffusion ($D_{b0}$) and bio-irrigation ($\alpha_0$) under varying depth-profiles of sediment mixing on $\Delta^{13}\mathrm{C}_{\mathrm{primary-bulk}}$.

## 4 Results

### 4.1 First-order temporal trends

By compiling $\delta^{13}\mathrm{C}_{\mathrm{carb}}$ data and unifying it in a biochronological framework, we can distinguish first-order temporal features of the combined Iranian and the Meishan P–Tr (sub)localities in China (Fig. 1). The subsampling routine prevents modifications by unequal sampling intensity, hence this temporal pattern is unbiased by the sampling strategy applied in individual studies. A gradual decline towards $4\,‰$ -lower $\delta^{13}\mathrm{C}_{\mathrm{carb}}$ can be discerned, starting from the middle Changhsingian, with minimum values ($0‰$) reached in the earliest Triassic, and a return to $2\,‰$ -higher $\delta^{13}\mathrm{C}_{\mathrm{carb}}$ commencing above the *I. isarcica* zone (unit I). Nonetheless, disparity in regional trends is visible as a double-peaked negative excursion marking the P–Tr transitional beds of Meishan, whereas such a signature is absent in time-equivalent boundary beds deposited at the Iranian sites.

By comparing combined subsampled median trend lines of individual studies (constructed following the same statistical routine, as outlined in Sect. 3.1.3) it is possible to analyse whether the first-order isotope trend is a consistent feature of the analysed stratigraphic sequences. A large portion of the compared individual datasets are marked with a high coefficient of determination ($r^2$) (Fig. 3). We can conclude that the geochemical signal obtained is reproducible and, as such, observable in rock collected during separate sampling campaigns over several decades. However, the comparative study also suggests that the correlation coefficients are consistently weaker in the Meishan data, relative to the Abadeh data. This lack of reproducibility might stem from limited stratigraphic coverage, e.g. Gruszczyński et al. (2003), which reports on a composite dataset of two sublocalities, Meishan D and Z, and only encompasses Changhsingian strata. Sampling campaigns of reduced stratigraphic

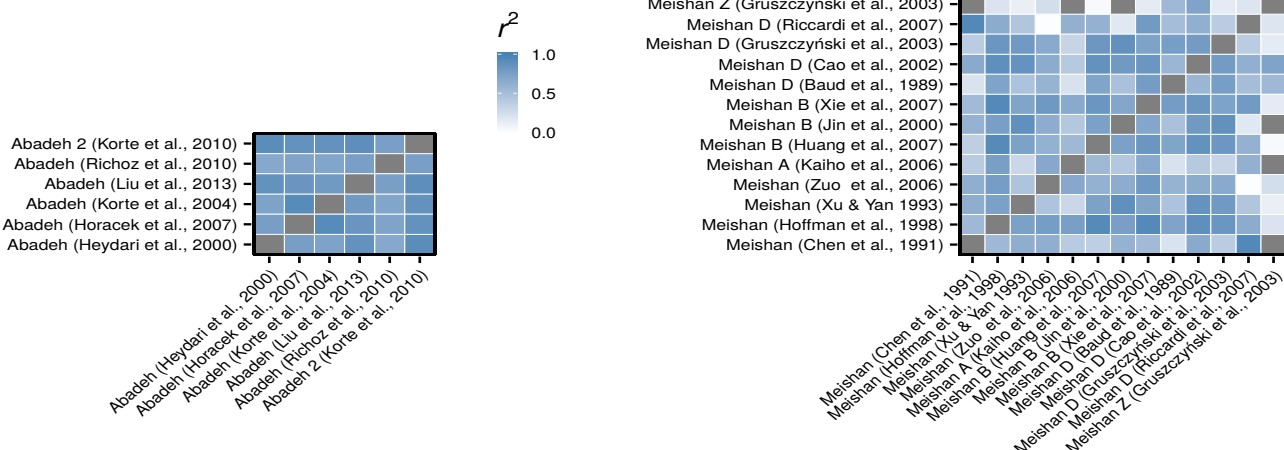

**Figure 3.** A comparative analysis of first-order temporal trends in $\delta^{13}C_{carb}$ from individual studies for the Abadeh and Meishan (sub)localities. The saturation-level of the individual tiles measures the correlative-strength of stratigraphic trends obtained during individual sampling campaigns, based on the coefficient of determination ($r^2$).

coverage might not be able to capture the first-order trend. Nevertheless, region-specific differences in signal reproducibility mark an overall greater disparity between individual Meishan $\delta^{13}C_{carb}$ records, and might point to a higher-order of $\delta^{13}C$ variability.

### 4.2 Residual carbon isotope variability

Subsampled and time-sliced median interpolations are depicted in Fig. 2. Trends with the highest confidence are highlighted by a well-defined white line, and the width of the CI is represented by a blue colour that becomes less intense with increasing data spread. Combined, these graphing features result in a more blurred image with a larger spread between the median values of individual subsampling routines (Sect. 3.1.3). From these figures we can discern three features in the second-order $\delta^{13}C$ excursions; 1) The residual $\delta^{13}C$ variability is seemingly random or stochastic and defined $\delta^{13}C$ excursions are unreproducible

across lithological successions from different geographic locations (Iran), or studies targeting the same site (e.g., Meishan section; Fig. 1). 2) A returning temporal pattern towards decreased fidelity of the median trend line (blurred white trend line) connected with increased CI dispersion (i.e., residual $\delta^{13}C$ variability, seen as less-saturated blue colour-tones), across the extinction horizon and ranging into the Early Triassic, at both geographic locations. Increased maximum $\delta^{13}C$ value-ranges from less than 1 and 2‰ (IQR), or 2 and 3‰ (IPR) in the pre-extinction beds, to more than 2 and 3‰ (IQR) or 5 and 8‰ (IPR)

around the extinction horizon, for Iran and China respectively, further enforces the observation of a globally significant peak in residual $\delta^{13}C$ variability. This pattern seems to recover after the P–Tr boundary, with generally higher confidence median trend lines and a trend to smaller maximum $\delta^{13}C$ value-ranges of less than 2 and 3‰ (IQR), or 5 and 8‰ (IPR) for Iran

and China, respectively. 3) In addition, a significant regional offset can be discerned in $\delta^{13}$C value-ranges, where the Chinese profile displays systematically higher residual $\delta^{13}$C variability.

Studies focussing on boundary events (e.g., the P–Tr transition) tend to channel sampling effort at a focal point around the presumed faunal turnover and horizon of palaeoenvironmental change (Wang et al., 2014). Sampling effort is known to have a profound impact on studies that use fossil data to track animal diversity through time (Alroy et al., 2008; Mayhew et al., 2012). By correcting for sampling effect through subsampling (Sect. 3.1.3), we cancelled out the potential effect of over-representative data accumulation on the temporal trend of stochastic residual $\delta^{13}$C$_{\mathrm{carb}}$ variability. Sampling effort might also have affected fossil collecting, thereby biasing conodont biozonation schemes, which heavily relies on the first appearance datum of certain fossil species (Wang et al., 2014; Brosse et al., 2016). As such, it would be perceivable that longer duration biozones could capture a more temporally variable marine DIC-$\delta^{13}$C, comparative to shorter duration biozones. When evaluating the average thickness and duration of our chosen biochronological units there is an apparent decrease in unit thickness, as well as shorter duration when approaching the extinction interval (Fig. 2). A comparison of biozone stratigraphic thickness and duration points to a relationship between higher $\delta^{13}$C$_{\mathrm{carb}}$ dispersion and smaller and shorter duration units (Fig. 2). This inverse relationship suggests that $\delta^{13}$C$_{\mathrm{carb}}$ variability is not controlled by the increased potential sample size and the inherent risk of sampling a more temporally variable isotope signature. Hence, we exclude the applied sampling strategy as well as the biochronological framework as a causal factor behind the observed temporal trend of stochastic $\delta^{13}$C$_{\mathrm{carb}}$ variability.

### 4.3   Model response to organic carbon accumulation

In the above, we have conceptualised a model that links sedimentation of organic material to carbonate carbon isotope alteration. An important variable in this model is the amount of organic carbon that arrives at the sea floor, which controls the redox depth profile and the respective importance of metabolic pathways, as reaction per unit area. Low organic carbon fluxes ($500\,\mu\mathrm{mol\,cm^{-3}y^{-1}}$) yield aerobic respiration as well as microbial-sulfate reduction as important biochemical reactions (Fig. 4). On the other hand, the importance of this metabolic pathway is reduced under high OC accumulation regimes ($>1000\,\mu\mathrm{mol\,cm^{-3}y^{-1}}$), which are signified by intense methane production by organoclastic methanogenesis (Fig. 4). Consequently, the dominant organic carbon remineralization pathway determines the final evolved carbon isotope composition of the equilibrated bulk-rock end member, e.g., towards lower $\delta^{13}$C$_{\mathrm{carb}}$ under low OC accumulation, and higher $\delta^{13}$C$_{\mathrm{carb}}$ under high OC loading of the sea floor (Figs 4–6).

By systematically changing $F_{OC}$ we see a defined relationship between bulk-rock C isotope alteration and the predominant mode of organic matter remineralization, e.g., by microbial sulfate reducers, AOM or methanogenesis (Fig. 5). The sensitivity experiments further reveal that the fraction of diagenetic precipitate incorporation, and bottom DIC have only a limited effect on the difference between the primary carbonate precipitate and the bulk-rock end member C isotope value ($\Delta^{13}$C$_{\mathrm{primary-bulk}}$). In contrast, marine sulfate levels modulate the total range of observed $\delta^{13}$C (~2–3.5‰) and introduce a switch in the system, which narrows the range of OC accumulation over which the maximum deviation in $\Delta^{13}$C$_{\mathrm{primary-bulk}}$ occurs (Fig. 5). In a similar fashion, elevated sedimentation ($v$ and $w$) causes a shift of peak $\Delta^{13}$C$_{\mathrm{primary-bulk}}$ values. How-

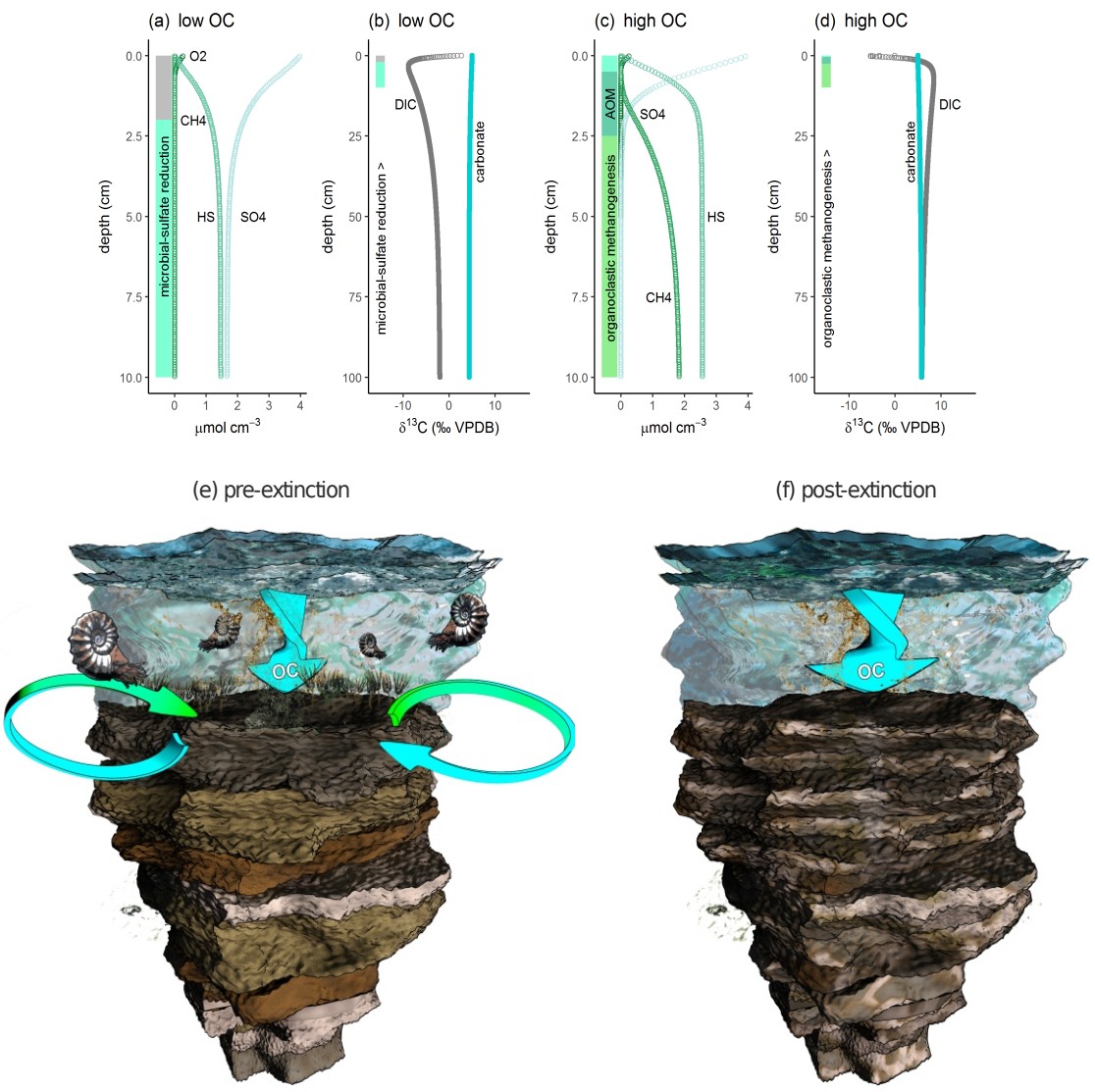

**Figure 4.** Diagenetic depth profiles of Late Permian sea floor sediments of a) oxidized and reduced solutes under a low shelf OC flux 500 $\mu$mol cm$^{-3}$y$^{-1}$, b) DIC and carbonate $\delta^{13}$C forced with a low shelf OC flux 500 $\mu$mol cm$^{-3}$y$^{-1}$, b) oxidized and reduced solutes under a high OC flux 1200 $\mu$mol cm$^{-3}$y$^{-1}$, d) DIC and carbonate $\delta^{13}$C forced with a high OC flux 7000 $\mu$mol cm$^{-3}$y$^{-1}$, e) situational sketch of pre-extinction bulk-carbonate accumulation with a normal OC flux and active benthic fauna and f) situational sketch of the post-extinction sedimentation with elevated OC accumulation, removal of metazoan benthic fauna and consequentially reduced sediment mixing (artwork by Mark Schobben; http://cyarco.com).

ever, this shift in peak values causes a modest increase of the total attained C isotope alteration with varying OC accumulation under higher sedimentation rates ($v$ and $w$).

A systematic study of OC accumulation results in comparable trajectories of diagenetic C isotope modification on bulk-rock by authigenic carbonate addition (Fig. 6). However, authigenic seafloor cementation results in a 10‰-range of attainable bulk-rock end member C isotope values, which is a wider range than obtained for diagenetic carbonate alteration. This style of carbonate cementation is also strongly controlled by seawater sulfate concentration. Lower than modern marine dissolved sulfate values ($28\,\mu\mathrm{mol\,cm^{-3}}$) allow for a large range of $\delta^{13}C$ end member values to be generated under a smaller range of $F_{OC}$ (Fig. 6). As opposed to diagenetic carbonate alteration, heightened sedimentation ($v$ and $w$) only causes a shift in in the switch from positive to negative $\Delta^{13}C_{\mathrm{primary-bulk}}$ in the end member rock but does not change the total range of C isotope alteration. Whereas, both elevated bottom water DIC and a decreased size of the authigenic fraction diminishes the maximum range of $\Delta^{13}C_{\mathrm{primary-bulk}}$.

Both bio-mixing and bio-irrigation by a pre-extinction Permian benthic fauna ($D_{b0} = 5\,\mathrm{cm^{2-}\,y^{-1}}$, $z_b = 2\,\mathrm{cm}$, $\alpha_0 = 50\,\mathrm{y^{-1}}$, $x_{irr} = 1\,\mathrm{cm}$) causes minimal modulation of $\Delta^{13}C_{\mathrm{primary-bulk}}$. The influence of modern benthic faunas ($D_{b0} > 5\,\mathrm{cm^{2-}\,y^{-1}}$, $z_b = 3\,\mathrm{cm}$, $\alpha_0 > 50\,\mathrm{y^{-1}}$, $x_{irr} = 2\,\mathrm{cm}$) would suppress $\delta^{13}C$ modifications by OC-steered diagenetic carbonate alteration and authigenic carbonate addition. On the other hand, the absence of a benthic fauna (post-extinction situation) allows for an unconstrained impact of previous cited OC-controlled trajectories of C isotope modification during bulk-rock formation and stabilization under P–Tr environmental conditions.

## 5   Simulation of a virtual carbon isotope time series

In order to understand the obtained temporal patterns in residual $\delta^{13}C_{\mathrm{carb}}$ variability, a series of reactive-transport models have been solved to steady state, under varying OC-sedimentation regimes. The virtual time series approach is an amalgamation of multiple sets of individual reactive-transport model runs (50 model iterations = 1 set = 1 time unit), sliding across a timeline of length 100, and with time increments of length one (Fig. 7).

By performing sensitivity test (Sect. 3.2.3), we have established that their exists a systematic relation between OC arriving at the seabed and the magnitude of bulk-rock C isotope alteration, expressed as $\Delta^{13}C_{\mathrm{primary-bulk}}$ (Fig. 5). Hence, the initial $F_{OC}$ can be estimated, because the $\Delta^{13}C_{\mathrm{primary-bulk}}$ can be approximated by the C isotope offset between calcite from well-preserved Permian brachiopods and time-equivalent bulk-rock samples (~ 1.0‰)(Schobben et al., 2016), and equates to $802\,\mu\mathrm{mol\,cm^{-3}\,y^{-1}}$. The $F_{OC}$ is subsequently modulated for each set (i.e., time unit) based on the observation that OC accumulation increases by a factor of four across the P–Tr transition (Algeo et al., 2013), and by linearly scaling this parameter to the observed residual $\delta^{13}C_{\mathrm{carb}}$ variability (IQR), as obtained from Fig. 2, yielding a continuous range for $F_{OC}$ ($802 \leq F_{OC} \leq 3206\,\mu\mathrm{mol\,cm^{-3}\,y^{-1}}$).

With regard to examining the effects of spatially distinct OC sinking fluxes and/or benthic fauna on the lateral distribution of OC , we forced individual model runs (within one time unit) with slightly deviating $F_{OC}$ values. The maximum attained $F_{OC}$ variability is constrained by adopting a log-normal skewed density distribution (R package *emdbook*; Bolker, 2008, 2016)

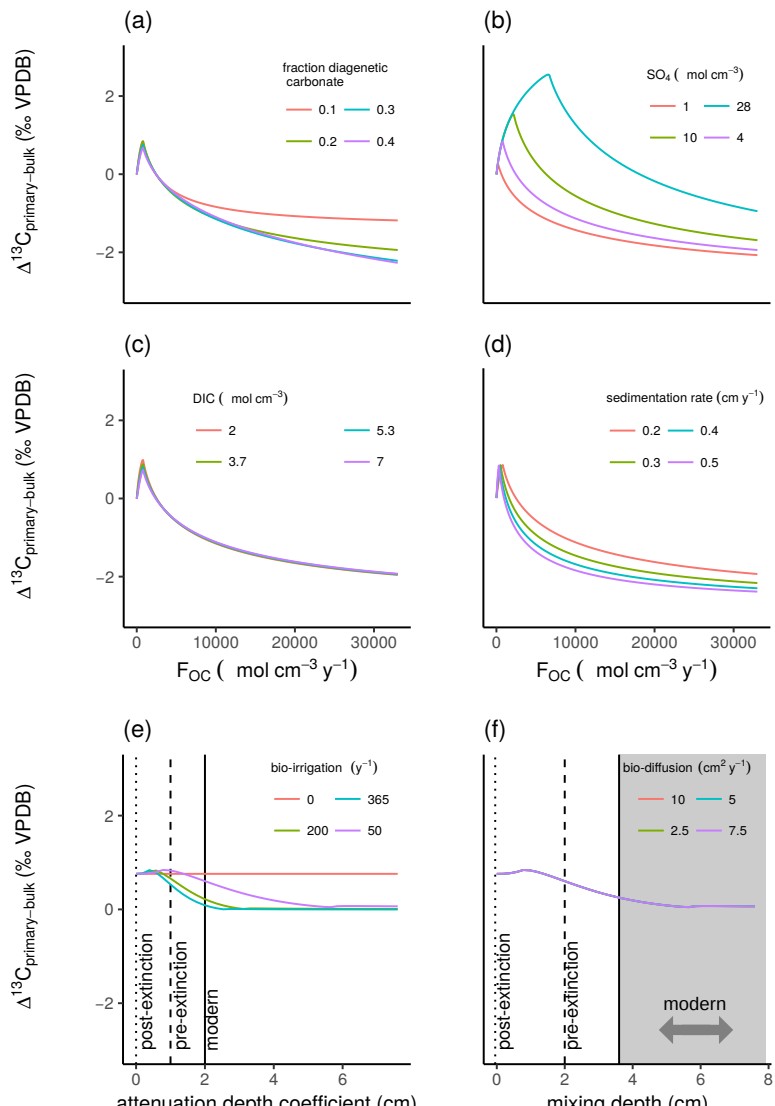

**Figure 5.** Sensitivity experiment for diagenetic carbonate alteration by equilibrium recrystallization with depth, designed to investigate the forcing effect of (a) the fraction of diagenetic carbonate incorporated ($f_{dia}$), (b) sulfate content of overlying water, (c) DIC content of overlying water and (d) sedimentation rate ($v$ and $w$), on the carbon isotope offset between the diagenetic end member rock and the primary calcite, over a range of $F_{OC}$. Panels (e) and (f) depict sensitivity experiments for bio-irrigation ($\alpha_0$) and bio-diffusion ($D_{b0}$), respectively, on diagenetic altered carbonate for changing modes of sediment reworking and irrigation by biota under a normal OC flux ($730\,\mu\mathrm{mol\,cm^{-3}y^{-1}}$). An increasingly suppressed alteration of the end member carbon isotope signal is observed for the pre-extinction (Dale et al., 2016) and modern (van de Velde and Meysman, 2016) depth profiles, compared with sediment that is not inhabited by metazoans (post-extinction).

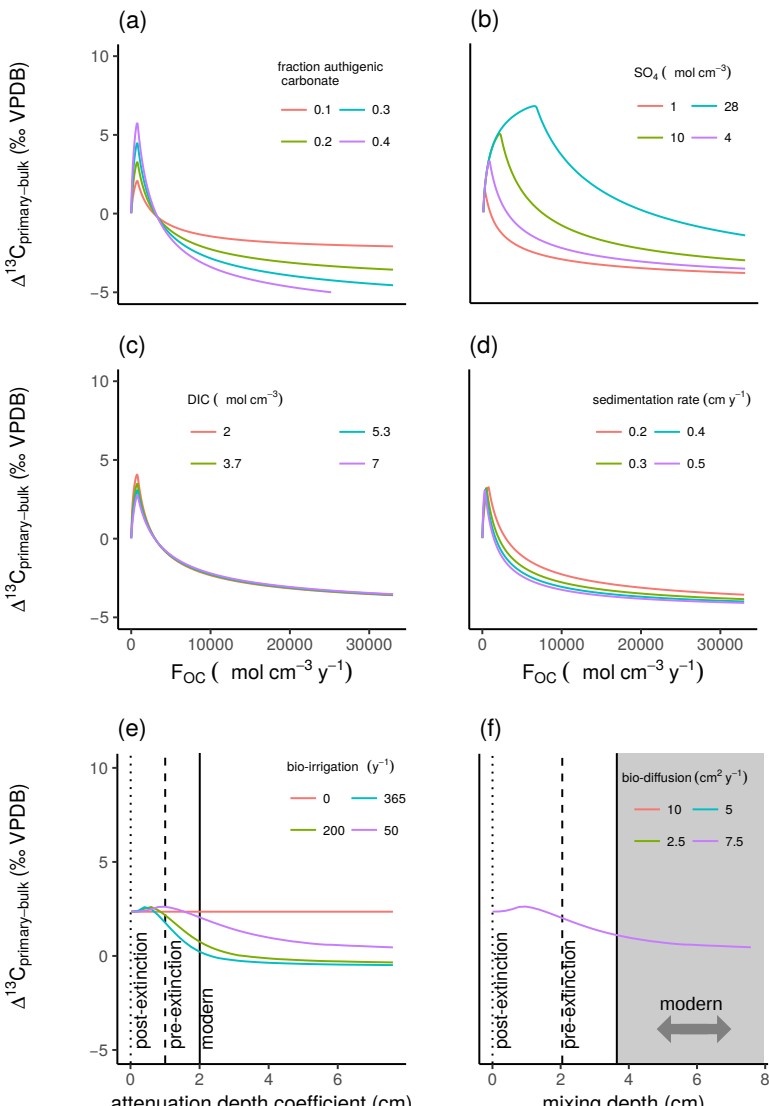

**Figure 6.** Sensitivity experiment for authigenic carbonate addition by seafloor cementation, designed to investigate the forcing effect of (a) the fraction of authigenic carbonate ($f_{auth}$) incorporated, (b) sulfate content of overlying water, (c) DIC content of overlying water and (d) sedimentation rate ($v$ and $w$), on the carbon isotope offset between the diagenetic end member rock and the primary calcite, over a range of $F_{OC}$. Panels (e) and (f) depict sensitivity experiments for bio-irrigation ($\alpha_0$) and bio-diffusion ($D_{b0}$), respectively, on authigenic carbonate addition for changing modes of sediment reworking and irrigation by biota under a normal OC flux ($730\,\mu\mathrm{mol\,cm^{-3}y^{-1}}$). An increasingly suppressed alteration of the end member carbon isotope signal is observed for the pre-extinction (Dale et al., 2016) and modern (van de Velde and Meysman, 2016) depth profiles, compared with sediment that is not inhabited by metazoans (post-extinction).

around the former established $F_{OC}$, which represents the population's median value (Fig. 7), and by randomly selecting a value from the created distribution. A log-normal skewed density distribution has been chosen to represent natural variation in the

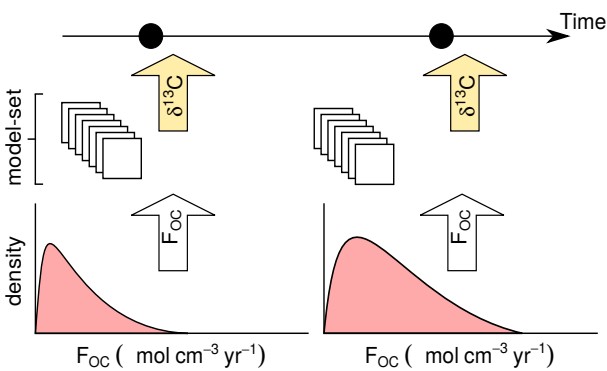

**Figure 7.** Schematic depiction of the work flow behind the virtual carbon isotope time series. Model sets consist of 50 separate reactive transport model solutions driven by a randomly selected $F_{OC}$. The randomly generated organic OC flux values translate into a density distribution with a right skewed tail. Increased $F_{OC}$ and a widening of the value-range is used to simulate increased spatial OC heterogeneity.

$F_{OC}$-population, which would be signified by variation that is skewed towards heavy sea floor OC loading in rare instances, whereas most variation is smaller in magnitude and can modulate the flux towards both smaller and higher contributions. Temporal variations in this parameter are attained by modulating the width of the sample population, ranging up to a factor of 1.5. This lateral OC-dispersion parameter is as well linearly scaled to the to the observed residual $\delta^{13}C_{carb}$ variability (IQR), as obtained from Fig. 2. This results in sample-population variations of $524 \leq$ median absolute deviation $\leq 3889\,\mu\mathrm{mol\,cm^{-3}\,y^{-1}}$, when combined with the initial $F_{OC}$-range, for homogeneous and more heterogeneous dispersed sedimentary OC, respectively.

The initial carbon isotope composition of $F_{carb}$ is based on the median $\delta^{13}C_{carb}$ values of the first time unit of the pooled Iranian dataset (Fig. 1). This value is subsequently corrected by $-1.0\text{‰}$, based on the systematic isotope offset between bulk-carbonate and pristine carbonate from brachiopod shells (Schobben et al., 2016), in order to obtain the C isotope value of $F_{carb}$. The systematic relationship of OC sedimentation with $\Delta^{13}C_{primary-bulk}$, as obtained from Fig. 5, yielded the primary $\delta^{13}C$ of calcium carbonate particles arriving at the seabed for successive time units.

The processes governing increased authigenic seafloor cementation in post-extinction strata are largely speculative. However, undisturbed substrates are a feature required to enable growth of calcifying microbial mats, regardless of the mechanism inducing the global-scale proliferation of these seafloor precipitates (e.g., Garcia-Pichel et al., 2004). Considering this universal control, it is justified to assume a direct link between bioturbation intensity and authigenic seafloor cementation. This assumption links both forms of carbonate rock formation and stabilization: equilibrium recrystallization and authigenic seafloor cementation. The probability ($p$) of producing authigenic seafloor cements has been assumed to vary linearly between: $0.01 \leq p \leq 0.1$, following the upper two tercile of the range attained by the previous defined OC-dispersion parameter. This definition determines the response of introducing authigenic seafloor cement $\delta^{13}C$ to the sample pool.

In order to further simulate the mass extinction of marine metazoans across the end-Permian extinction on sedimentary conditions, the parameters that define bio-mixing and bio-irrigation (Sect. 3.2.2) are set to zero from time unit 60 onward (equivalent to biozone G; Fig. 1 and Table 1). In addition, it has been postulated that sedimentation rates increased over the

P-Tr transition (Algeo and Twitchett, 2010), so the sedimentation rate (of $v$ and $w$) was modulated at time unit 70 (equivalent to biozone H; Fig. 1 and Table 1). However, a conservative two times sedimentation increase is adopted, as the previous cited estimates largely hinges on calculations based on the duration of the respective stages, which are subject to continues modification (Li et al., 2016). The $f_{dia}/f_{auth}$ for both trajectories of carbonate rock stabilization is kept constant at 0.2. The final bulk-rock carbon isotope value is obtained by mass balance from the calculated primary and diagenetic/authigenic carbonate components.

A total of 5,050 reactive-transport models have been solved in order to match the length and resolution of the cumulative "real" $\delta^{13}$C time sequence from sites in Iran (Fig. 1). The resulting virtual time series of model runs forced by previously established value-ranges for the OC sinking flux, sediment homogeneity and authigenic mineral addition is depicted in Fig. 8. Although these steady state solutions will not always be truly representative for a dynamic sedimentary environment (e.g., sedimentation rate changes and sediment reworking), we conclude that the conceptualised model (Sect. 2) and selected parameter values can explain trends in residual carbon isotope variability.

# 6 Discussion

## 6.1 Stochastic carbon isotope signatures of the Permian–Triassic boundary beds

Global comparisons of P–Tr carbon isotope records have uncovered disparate $\delta^{13}$C trends and excursions (Fig. 1). These deviations are often encountered on refined stratigraphic and lateral scales, complicating interregional high-resolution correlations (Heydari et al., 2001; Hermann et al., 2010). Subaerial exposure and projected trajectories of bulk-carbonate stabilization under the influence of meteoric fluids and high water–rock ratios might be invoked to explain this disparity (Heydari et al., 2001; Li and Jones, 2017). Other studies have pointed to stratigraphic variations in predominant mineral rock composition as a source of $\delta^{13}$C$_{carb}$ modulations, citing the mineral-specific isotope offset between aragonite, dolomite and low-Mg calcite (Brand et al., 2012b; Heydari et al., 2013; Schobben et al., 2016; Li and Jones, 2017).

In this study, we provide an alternative explanation for disparate second-order $\delta^{13}$C$_{carb}$ excursions, involving the observed stochastic residual $\delta^{13}$C variability at a refined spatial scale (Fig. 1). We regard this test to be of importance, as a transgression marks the boundary beds and Lower Triassic deposits (Fig. 1). Furthermore, subaerial exposure and consequential early rock stabilization with high water to rock ratios (open system) of the studied localities is unlikely to have left an isotopic overprint. Under such circumstances regionally extensive carbonate sediment dissolution by contact with an undersaturated solution could have sourced the underlying sediment pile with saturated fluids for cementing (Bathurst, 1975). However, evidence of such exposure in the form of karstification is absent in the studied locations (Leda et al., 2014; Yin et al., 2014). A recent study on the Meishan section suggest that recrystallization (i.e., zoned dolomite crystals) and modification of C isotope composition of certain stratigraphic levels was forced by a short-lived regression and consequential exposure to meteoric water (Li and Jones, 2017). However, the assignment of such petrological indices to specific diagenetic environments is often fraud with uncertainties, and it is not uncommon that alleged diagnostic features occur in a range of diagenetic environments (Melim et al., 2002). Although a complete exclusion of isotope resetting by meteoric water is not possible, it is possible that a large

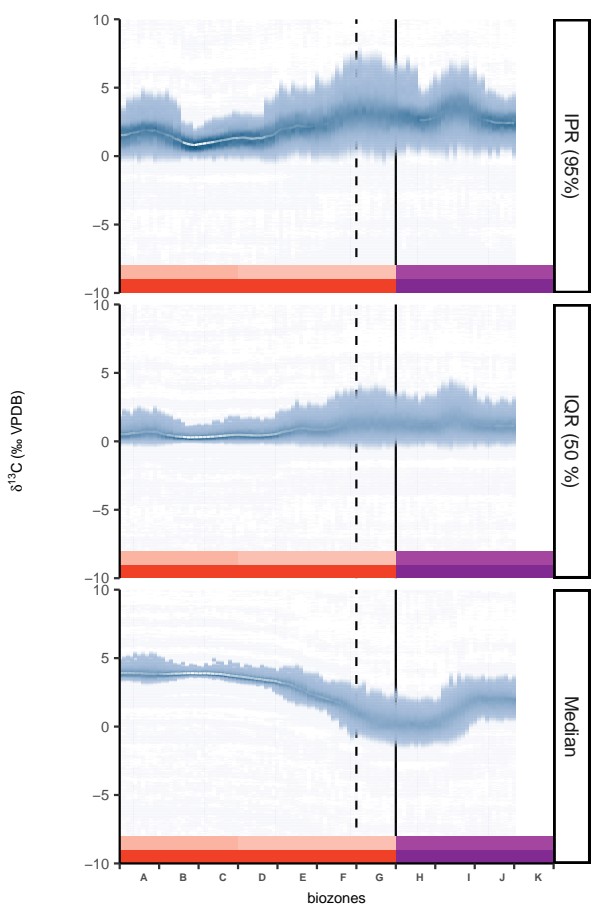

**Figure 8.** Subsampled virtual carbon isotope time series with model results plotted as "visually weighted" trends. Each time-unit incorporates 100 model iterations with a randomly chosen OC flux from a log-normal skewed density distribution. The time series simulates an increase of four times the average OC flux, an 1.5 times increase in the spatial-heterogeneity of OC accumulation, and increased likelihood of producing seafloor authigenic carbonate to a maximum of 1 in 10. Bioturbation (expressed as bio-mixing and bio-irrigation) is modulated at the extinction horizon to mimic the effect of a die-off among benthic organisms. In addition, sedimentation ($v$ and $w$) is increased twofold over the P-Tr transition (see main text for sources).

portion of the global carbonate archive lithified without being subjected to meteoric fluids, therefore justifying the exploration of alternative modes of isotope resetting.

Systematically [12]C-depleted bulk-rock relative to brachiopod calcite (Schobben et al., 2016) furthermore counters the notion of a variable predominance of an aragonite precursor for Permian carbonate-rock (Brand et al., 2012b; Heydari et al., 2013). Nonetheless, secondary calcite and dolomite additions might be a causal factor behind observed $\delta^{13}$C variability. However, these additions can also be the result of *in situ* microbially steered mineral formation, which incorporates metabolically produced DIC isotope signatures (Bontognali et al., 2014; Li and Jones, 2017). As such, our conceptualized model is not mutually

exclusive with regards to carbonate polymorph compositional changes, but it does not require rapid global secular marine ion inventory changes. Moreover, our model outcome predicts $\delta^{13}C$ variation driven solely by small biologically steered mineral contributions ($f_{dia}/f_{auth} = 0.2$)(Fig. 6), whereas the recorded $\delta^{13}C$ fluctuations are up to 10 times higher in amplitude than otherwise predicted for thermodynamically driven mineral-specific isotope compositional offsets (cf. Brand et al., 2012b; Heydari et al., 2013). Note also that the mineral-specific isotope offset would require bulk-rock samples to display effectively complete (100 %) mineral compositional changes to a pure carbonate-polymorph composition. In contrast, our model predicts a mechanism that can introduce temporal variation in spatially heterogeneous carbonate polymorph assemblies and associated isotope compositional differences by post-depositional processes, driven by spatially and temporally variable OC accumulation comparable to modern ranges (cf. Müller and Suess, 1979). Depleted marine dissolved sulfate concentrations would have exacerbated the sensitivity of carbonate rock towards only small deviations of this allochthonous C-source (Figs 5 and 6). Combined with a less well-mixed sediment and microbially steered cementation this model can account for the entirety of the observed residual $\delta^{13}C_{carb}$ variability.

## 6.2   The carbon isotopic composition of the Permian–Triassic DIC reservoir

The first-order P–Tr negative carbon isotope trend can be explained by changes in the sources and sinks of the long-term (> 100 ky) carbon cycle, e.g., a reorganisation of organic carbon burial and volcanic $CO_2$-outgassing (Renne et al., 1995; Kump and Arthur, 1999; Berner, 2002). Carbon-cycle partitioning between the deep and shallow ocean through the marine biological pump have been proposed to explain the depth-gradient isotope differences (Meyer et al., 2011; Song et al., 2013) or a rapid (< 100 ky) C isotope excursion (Rampino and Caldeira, 2005). On the one hand, the stochastic residual $\delta^{13}C_{carb}$ variability, at confined cm–m stratigraphic-scale, is inherently difficult to reconcile with transient perturbations of the dynamic C-cycle equilibrium enforced by repeated biological reordering of DIC between the shallow and deep ocean (i.e., consecutive events of global productivity increase and wholesale collapse). On the other hand, if lateral variations in marine DIC-C isotope composition accounts for $\delta^{13}C_{carb}$ variability, water column mixing would likely erase spatial compositional differences on smaller scales (< km-scale). For example, residual $\delta^{13}C_{carb}$ variability recorded at Meishan (Fig. 1), at a cm to m-scale, cannot be reconciled with heterogeneity of the overlying watermass, and less well-mixed sediments becomes a more acceptable alternative. Although geographically disparate C isotope signals of different sites in Iran (> km-scale) could instead be accounted for by heterogeneity of watermasses, the temporal trend in the amplitude of the residual $\delta^{13}C_{carb}$ variability is similar for both China and Iran (Fig. 2). An overarching control of the here-invoked OC-steered diagenetic mechanism is the most parsimonious model to explain this geographical coherence.

Seafloor methane clathrate dissociation, coal and organic-rich sediment combustion by igneous intrusions as well as ocean Ni-fertilization stimulating methane production by methanogens are viable sources of transient (< 100 ky) isotope depleted carbon contributions to the atmosphere–ocean system (Erwin, 1993; Knoll et al., 1996; Krull and Retallack, 2000; Retallack and Jahren, 2008; Svensen et al., 2009; Rothman et al., 2014). Although all of the cited mechanisms are conceivable triggers for defined second-order $\delta^{13}C_{carb}$ excursions, they are less likely to create spatially divergent isotope trends and repeated temporally distinct fluctuations, and instead essentially reflect bed-to-bed variation. The existence of transient carbon isotope

excursions is also unlikely, considering the predicted elevated DIC levels of an ocean without pelagic calcifiers, resulting in a conservative behaviour of the DIC-pool towards transient C isotope perturbations (i.e., increasing the systems response time; Kump and Arthur, 1999; Zeebe and Westbroek, 2003; Rampino and Caldeira, 2005). Nonetheless, second-order (transient and depth-gradient) C isotope signals of a primary origin might still be imprinted in the isotope records. However, rock formed

under the previously outlined marine depositional conditions (with intensified spatially heterogeneous trajectories of C isotope alteration), increases the chance of sampling spatially variable and diagenetically modulated $\delta^{13}C$ signals, and these would obscure primary signals.

## 6.3 Perspectives on the Permian–Triassic carbonate archive: stratigraphy, biological evolution and the global biogeochemical carbon cycle

These findings put renewed constraints on the application of whole-rock $\delta^{13}C$ as a high-resolution stratigraphic tool. Diagenesis forced by a recrystallization process, under the influence of meteoric fluids, high fluid to rock ratios, and oxidized terrestrial-derived organics, is generally considered to be the main driver of bulk-rock C isotope alteration (Brand and Veizer, 1981; Marshall, 1992; Brand et al., 2012a, b). Rocks without petrological evidence diagnostic of meteoric fluids percolating through voids and interacting with sediment grains, e.g., palaeokarsts, blocky calcite and pendant cements, are usually regarded

as pristine and minimally isotopically altered (Veizer et al., 1999; Flügel, 2010; Brand et al., 2012a, b). Regarding the observed P–Tr residual whole-rock $\delta^{13}C$ variability and model predictions, we have shown that under certain marine and sedimentary situations marine carbonate rock diagenesis can introduce significant $\delta^{13}C$ fluctuations to the whole-rock isotope composition. The insight gained here undermines the idea of using second-order $\delta^{13}C$ fluctuations as stratigraphic markers, which are often confined to limited stratigraphic intervals (e.g., individual limestone beds), without a proper understanding of the potential ef-

fects of diagenetic carbon isotope modification. At least for the data presented here, we can demonstrate that shifts in bulk-rock $\delta^{13}C$ smaller than classical biozonation schemes have to be considered with caution, when they coincide with sedimentological features of reduced bioturbation (e.g., lamination), and likely do not serve as a globally universal marker. Instead, long-term first-order trends, cross-cutting formational boundaries (Supplementary Information), have a higher likelihood of representing secular variation in seawater DIC isotope composition, and are largely unaffected by marine diagenetic processes (Fig. 1).

Extrapolation of our model results to Mesozoic and Cenozoic bulk-carbonate records signified by comparatively reduced $\delta^{13}C$ variability might reflect buffering of highly variable diagenetic C isotope contributions under elevated marine sulfate levels and physical sediment mixing by benthic fauna.

The carbonate chemical signal towards increased $\delta^{13}C$ variability supports the global significance of three parameters during deposition of P–Tr carbonate rock: reduced sediment mixing, authigenic precipitates and heterogeneous OC accumulation.

Although we have postulated a connection between sediment lamination and C isotope variability, the relation of these observations to the end-Permian mass extinction is unconstrained. The cataclysmic mechanism that drove the extinction of benthic faunas is still a matter of debate, but bottom water anoxia is one of the main contenders. This could have been caused by decreased seawater oxygenation instigated by stagnating ocean circulation (Wignall and Twitchett, 1996; Winguth et al., 2015). Alternatively, a higher OC sinking flux might have led to increased $O_2$ drawdown by aerobic respiration and $H_2S$ produc-

tion by sulfate-reducing microbes, leading to widespread marine euxinia (Meyer et al., 2008; Algeo et al., 2013; Schobben et al., 2015). However, $\delta^{13}$C variability complies most compellingly with at least continuing OC accumulation over the studied interval, and thereby largely undermines scenarios of reduced primary productivity (e.g., Rampino and Caldeira, 2005). On the other hand, the proliferation of post-extinction authigenic seafloor precipitates (e.g., thrombolites, stromatolites and fan-shaped structures) likely represents a symptom of reduced sediment disturbance and elevated carbonate saturation (cf. Kershaw et al., 2009). Since some of the more conspicuous post-extinction seafloor structures have been connected with microbial communities and distinct metabolism-mediated isotope signatures (e.g., Heindel et al., 2013), they might be equally important constituents of the observed spatial $\delta^{13}$C variability. Geographic differences observed as systematically higher residual $\delta^{13}$C variability at the Chinese localities are suggestive of a regionally distinct hydrographic and depositional setting. This regional feature could be linked with evidence of episodic Late Permian euxinic conditions in the South China basin (Grice et al., 2005; Cao et al., 2009). Comparatively higher, and spatially variable, OC loading of the sea floor in the South China basin under these circumstances, might be triggered by enhanced primary productivity or OC sinking fluxes (Algeo et al., 2011, 2013). On the other hand, the depositional sites found in Iran lack evidence for bottom water anoxia and favour other drivers behind the marine faunal extinction, e.g. thermal stress or ocean acidification (Leda et al., 2014; Schobben et al., 2014; Clarkson et al., 2015).

Besides capturing regional biological expressions of the global faunal disruption, our approach illuminates potentially important feedbacks of the Earth system that control the global-scale carbon cycle. Recently, attention has increased on the potential significance of authigenic carbonate production on the global carbonate reservoir (Schrag et al., 2013), and the addition of authigenic carbonate to Permian and Triassic carbonate-bearing sequences in particular (Schobben et al., 2016; Zhao et al., 2016). The numerical exercise highlights the link between local sedimentary OC cycling and bulk-rock diagenetic stabilization, hinting at the potentially significant portion of remineralized OC sequestered by authigenic carbonates. Additionally, bulk-rock residual $\delta^{13}$C variability might serve as an indicator of infaunal biological activity, effectively preserving an ecological signal, which could, in turn, be an indicator of local sedimentary C-cycling. This sub-cycle is strongly controlled by bioturbation and controls the amount of buried organic matter as well as the supply of electron acceptors (Canfield, 1989; Canfield et al., 1993). These fluxes drive carbon removal from the exogenic carbon reservoir and potentially require a revision of generally accepted constraints on these budgets, including during events with postulated major C-cycle perturbations, such as the latest Permian extinction.

## 7   Conclusions

The $\delta^{13}$C$_{carb}$ record straddling the P–Tr boundary of Iran and South China can be decomposed in a first-order temporal trend, with a negative trend marking the transition beds, and superimposed second-order residual $\delta^{13}$C variability. The primary goal of the current study was to delineate the nature of the second-order $\delta^{13}$C fluctuations, which harbour a reproducible temporal pattern towards increased variability across the extinction beds. By investigating the origin of these second-order geochemical signals it is possible to better constrain the limits of carbon isotope chemostratigraphy, most notably the time resolution on

which it can be applied. Having established that second-order C isotope variability traces local effects rather then the global carbon reservoir, we discerned that mineralogical heterogeneity as well as meteoric-steered cementation are unsatisfactory candidates to explain this spread in $\delta^{13}$C. Hence, the possibility of metabolism-steered carbonate addition is more likely, noting that the anaerobic microbial pathway introduces isotopically depleted (or enriched) carbon to the porewater, and that the same

microbes are often connected with production of diagenetic and authigenic carbonates. Moreover, organic carbon degradation occurs in a part of the sediment column signified by a high reactivity of the primary carbonate sediments. By testing these assumptions with reactive-transport models, and with the construction of a virtual time series, we can recreate the observed variability in the end member bulk-rock $\delta^{13}$C composition by inducing heterogeneous spatial-distribution of sedimentary OC, i.e., a reduction of physical sediment mixing or spatially divergent OC sinking fluxes. Authigenic seafloor cementation,

possibly mediated by microbial mat communities, is another source that might have introduced diagenetic overprints in the P–Tr boundary beds. On the other hand, these mechanisms can still preserve long-term trends of oceanic DIC-carbon isotope composition. These findings strengthen the notion that bulk-carbonate $\delta^{13}$C is a fruitful source of information, recording secular trends in carbon cycle evolution as well as potentially informing us about the biosphere, notably infaunal animal activity, seafloor microbial communities and OC delivery to the seafloor.

**8   Code availability**

Data visualization, statistical data treatment as well as the reactive-transport model have been written in the R-language. The R-scripts are made available on the GitHub repository: https://github.com/MartinSchobben/carbonate (DOI: 10.5281/zenodo.888754).

**9   Data availability**

The combined data-mined and newly produced $\delta^{13}$C are made available on the GitHub repository: https://github.com/MartinSchobben/carbonate (DOI: 10.5281/zenodo.888754).

*Author contributions.*   MS designed the study. MS, CK, VH, AG, LL collected material in the field and prepared samples for isotope analysis. US, CK and CVU performed C isotope analysis. MS and SvdV constructed the numerical model. MS and JS data-mined complementary published C isotope results. All authors provided intellectual input and contributed to writing the manuscript.

*Competing interests.*   The authors declare that they have no conflict of interest.

*Acknowledgements.*   We thank Sylvain Richoz (University of Graz, Austria) and Yoshitaka Kakuwa (University of Tokyo, Japan) for providing us with formattable tables of published carbon isotope data. We acknowledge support provided by Aras Free Zone Office and Adel

Najafzadeh (Tabriz) to sample the Ali Bashi, Aras Valley and Zal sites, and Bo Petersen (University of Copenhagen) for analytical assistance in the laboratory. This project was funded by Deutsche Forschungsgemeinschaft (DFG; projects KO1829/12-1, KO1829/12-2 and KO2011/8-1). MS is currently funded by a DFG Research Fellowship (SCHO 1689/1-1), SVDV is supported by a Ph.D. fellowship from the Research Foundation Flanders (FWO), CVU acknowledges funding from NERC grant NE/N018508/1, and SWP acknowledges support from a Royal Society Wolfson Research Merit Award. The publication of this article was funded by the Open Access Fund of the Leibniz Association.

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
