# Peer review of "Latest Permian carbonate-carbon isotope variability traces heterogeneous organic carbon accumulation and authigenic carbonate formation"

_Climate of the Past, 2017_

## Referee Comment (RC1) · Anonymous Referee #1 · 7 Jun 2017

The authors explain the scatter of d13C near the P-T boundary as recrystalizaion and the formation of authigenic carbonates during marine diagenesis. They also use a reaction-transport model to reproduce the d13C data. While this idea is new and the disscussion is great, there are some major issues with this manuscript.

The authors try to interpert the scatter in the d13C records as recrystalization with an organic carbon source during marine diagenesis. However, it is also possible that some of these variations may have been generated by meteoric diagenesis. Thus, more disscussions on this point are required, and it's better to include some petrography or

geochemical evidences. The variations may also be generated by changes of rock types, minerals and calcified fossil species. Thus, more descriptions on the samples are necessary.

The data complied from the Meishan section in South China maybe can preclude the influence of seawater chemistry. However, the Iran data do not come from the same site. The scatter in d13C may be generated by spatial heterogeneity in seawater chemistry rather than diagenesis.

The authors have talked about authigenic carbonates. A definition on authigenic carbonates is required. If the carbonates were mainly formed by recrystalizaion, their d13C value may have been changed due to the exchange of carbonate ions with pore-water. Can this type of carbonates be classfied into authigenic carbonates? Except some shells, all ancient carbonates have suffered from recrystalization. Does this means that all the ancient carbonates are authigenic carbonates?

Some further textual suggestions:

Page 2, line 13 methanogenesis belongs to degradation of organic matter.

Page 5, section 3.1.2. Some of the d13C data for the Meishan section are very old and are less -5‰ which may be generated by analytical error.

Page 7, equation (1), why there is no a concentration conversion factor $(1-\varphi)/\varphi$ in the reaction term? equation (2), I guess Db(z) is biodiffusion here, should describe it and also write down the function.

Table 2. I don't know why SO4/(SO4+KSO4) is in the reaction rate law of anaerobic oxidation of methane.

Page 9, line 15. The sedimentary rate may have been changed across the P-T boundary.

Page 14, line 15. "This inverse relationship suggests that d13Ccarb variability is not

controlled by the increased potential sample size". This doesn't make sense to me. If there is more sampling effort on short interval, it is possible to capture larger d13C variablity. Could you show the relationship between the d13C variablities and the numbers of data of different intervals?

Figure 5 is good. Could you also show the influence of biodiffusion? The intensity of biodiffusion could be a function of oxygen level. It is better if bioirrigation is also included.

Page 22. The discussions about the influence of seawater chemistry and meteoric diagenesis are great. It may be better to talk about the other explanations for the scatter d13C such as heterogeneity in seawater chemistry, meteoric diagenesis and the variation of mineralogy in one section. Also, all the other disscussions can be putted into another section. These may make the structure of the disscussion part more clear.

---

## Referee Comment (RC2) · Anonymous Referee #2 · 7 Jul 2017

Review of CP-2017-66 Latest Permian carbonate-carbon isotope variability traces heterogeneour organic carbon accumulation and authigenic carbonate formation

Schobben et al. analyze a large suite of new and existing carbon isotope data from the P/T boundary of Meishan, China, and Abadeh, Iran. Using a subsampling routine over a timeline normalized by biozones, the authors determine median first order trends at each location and the variance around the median. The authors then utilize a diagenetic reactive-transport model to probe the potential for authigenic carbonate precipitation and recrystallization to push bulk d13C values of carbonate sediments

away from primary values, creating secondary trends in d13C that do not or indirectly reflect ancient seawater conditions. The authors suggest that higher and more spatially variable fluxes of organic carbon, as well as lower sulfate concentrations, during the P/T event lead to greater variance of bulk d13C values due to greater production of authigenic carbonate.

Overall, this is a great study that will impact the P/T paleobiology community, as well as the broader isotope geochemistry community. I am excited to see the authors take a rigorous, statistical approach to their isotope data, and embrace the "nuanced view on the nature of bulk-rock carbon isotope composition...where pure primary and strictly diagenetic end member states are considered as a continuum." Generally, the paper is well-written, though at times the language is a bit dense and hard to follow. I have tried to outline some examples below.

I have two primary concerns for this paper, both related to the treatment of carbonate chemistry. First, how does the diagenetic (reactive transport) model account for changes in the carbonate system? It seems that the model considers CO2 aqueous, bicarbonate, and DIC, but this is incomplete. Bicarbonate/CO2 additions and subtractions to the DIC pool do not simply add/subtract to those supplies. Additions of those species will cause re-equilibrium among all DIC species. These re-equilibrations will results in non-linear changes to calcite saturation.

On a related note, I don't understand the kinetic rate expressions that are provided, and the explanations are insufficient. The authors cite Boudreau 1997—I checked the reference and could not find an explanation, though it could be because the reference is a ∼400 pg book. Why are all of the rate expression factors of O2 or SO42-? How does a monod constant compare to a solubility product or rate constant? Obviously I am not as well versed in these things as the authors of this paper, but I asked two other scientists in related fields and they were similarly confused. The authors must make these concepts more accessible to their target audience. Citing a 400 pg book is not sufficient.

Second, the authors assume constant boundary conditions for their model, but also recognize changes in the carbon isotope composition of seawater DIC as a result of "changes in the sources and sinks of the long-term (>100 ky) carbon cycle". The changes in the carbon cycle are also affecting changes in carbonate chemistry, which should affect the ability of authigenic carbonate to precipitate and how much dissolution/equilibration takes place. See Payne et al. 2010 in PNAS to start.

Below are some minor comments I have on the text and figures: —The authors often use the term "sedimentation rate", which I take to mean F_carb? They do not seem to include F_OC under this definition? Regardless, the use of "sedimentation rate" is confusing and should be made more specific—perhaps "carbonate sedimentation rate" or "calcite sedimentation rate".

—Pg. 3, line 22: What do the authors mean by "High carbonate ion concentrations"? Are we talking about DIC or CO32-? If these high values are predicted, why did the authors use modern DIC values in their model?

—Pg. 3, line 26: "High carbonate ion concentrations are invoked..." True, and this implies a change in seawater carbonate chemistry that should be considered in the time series diagenetic model.

—Pg. 4, lines 7–15: The phrases in italtics are confusing. I am having a hard time parsing their meaning and tying them to the clauses before them. Are they even used later? I would just get rid of them.

—Pg. 4, lines 16: should be "studied". "To carry out this investigation" implies action in the past.

—Pg. 6, lines 3–13: Can the authors reference a figure here? It would be easier to follow the explanation with a visual aid.

—Section 3.2.1: Have the authors considered the role of other metabolisms, such as Fe and Mn reduction? These metabolism can yield much more alkalinity than sulfate

reduction (see Bergmann et al. 2013 in Palaios). I suspect that Fe reduction is not quantitatively important for most of the Phanerozoic when sulfate and O2 are high, but if sulfate and O2 is low at the P/T it could be significant.

—Pg. 10, lines 10–12: "The previous...solid carbonate." This sentence is hard to follow. Please consider clarifying.

—Figure 1: The median line is completely obscured for the Iran plot. Can the authors move the line to in front of the data points?

—Pg. 14, line 27: "depleted" and "heavier" are incorrect terms. Should be "lower" and "higher". From Sharp 2007 (Principle of Stable Isotope Geochemistry pg. 16): "As numbers, delta values can be high or low, positive or negative, but not heavy or light, [nor can they be]...depleted or enriched."

—Figure 3: What is the range of values represented by the green colors in the biozone thickness/duration graphs? Does more saturated mean shorter or longer?

—Figure 7: I don't understand the point of this figure. I understand that the authors are changing the distributions from which F_OC is generated in the model, but the rest of the figure is lost on me.

—Pg. 19, line 5: Can the authors elaborate on how D13C_primary-bulk was derived in Schobben et al. 2016? It would be useful to do here, so the reader can understand this paper without first reading the other.

---

## Author Comment (AC1) · 31 Jul 2017

We would like to thank the reviewer for the constructive comments on the manuscript. We provide a point-by-point reply to concerns prompted by the reviewer. In some cases we agree with the reviewer and in other cases we do not. In the latter case we give a full explanation as to why we regard the criticism as unfounded.

Comment #1: The authors try to interpret the scatter in the d13C records as recrystallization with an organic carbon source during marine diagenesis. However, it is also

possible that some of these variations may have been generated by meteoric diagenesis. Thus, more discussions on this point are required, and it's better to include some petrography or geochemical evidences. The variations may also be generated by changes of rock type, minerals and calcified fossil species. Thus more descriptions on the samples are necessary.

Response #1: We of course agree that lithology, diagenesis (including that by meteoric waters), polymorphism and polymorph-specific C isotope values can alter C isotope values. However, we point out that our conceptual model is an exercise to show that diagenetic pathways, fuelled by marine organic matter, can produce the observed patterns. This does not necessarily mean that the real variation is - in all instances - caused by this mechanism, but we merely point to its excellent potential to explain the magnitude of the variations and the temporal pattern of variability across the P–Tr boundary.

In addition, a large suite of petrological work has already been performed on most of the studied sites (referred to in the original work and the online supplementary text). These studies point to the unlikeliness of diagenetic alteration of these particular suites of rock by the interaction with meteoric fluids. We further consider it unlikely that polymorphic variations can account for some of the larger $\delta$13C fluctuations (Section 6.1).

If we included a petrological study within the main text, it would substantially increase the length of an already long text and detract from the overall message. We will, however, stress in the revised work that the model serves as a new concept and an alternative to classical views on carbonate rock diagenesis.

Comment #2: The data compiled from the Meishan section in South China maybe can preclude the influence of seawater chemistry. However, the Iran data do not come from the same site. The scatter in d13C may be generated by spatial heterogeneity in seawater chemistry rather than diagenesis.

Response #2: We agree with this possibility, as stated in lines 12 to 14 on page 22.

However, trends in residual $\delta$13C variability from both regions show the same temporal pattern (Figure 3), suggesting that the postulated diagenetic mechanism is of global significance. This prevents the interpretation of variations as observed among different sites of Iran in terms of heterogeneity in seawater chemistry, as it might just as well relate to the described diagenetic mechanism. Additional discussion will be added in the revised work.

Comment #3: The authors have also talked about authigenic carbonates. A definition on authigenic carbonates is required. If the carbonates were mainly formed by recrystallization, their d13C value may have been changed due to the exchange of carbonate ions with pore-water. Can this type of carbonates be classified into authigenic carbonates? Except some shells, all ancient carbonates have suffered from recrystallization. Does this means that all the ancient carbonates are authigenic carbonates?

Response #3: Mixing of dissolved carbonate and metabolic-derived carbon is accounted for in the current model. This is visually expressed in Figure 4b,d and mathematically expressed in equation (4) which accounts for the rate of dissolution and crystallization. Since dissolution adds new DIC to the porewaters, this constant exchange somewhat buffers the metabolic-derived carbon signal

However, the reviewer is correct in expressing concerns about the original definitions where "authigenic carbonate" included equilibrium recrystallization that occurs over a range of sediment depths and microbially-mediated carbonate cementation (mostly occurring at the sediment-water interface). The former mechanism comprises relatively buffered carbonate and is therefore probably better defined as a diagenetically altered carbonate rock. in which the carbonate has partially dissolved and been replaced by a less reactive phase. We will include an updated definition in the revised manuscript, where we divide between diagenetically altered carbonates and authigenic carbonate addition.

Comment #4: Page 2, line 13 methanogenesis belongs to degradation of organic matter.

Response #4: Methanogenesis will be omitted.

Comment #5: Page 5, section 3.1.2. Some of the d13C data for the Meishan section are very old and are less -5‰ which may be generated by analytical error.

Response #5: The precision of carbonate carbon isotope measurements has not changed over the last decades, and therefore does not introduce an increasing measurement bias with progressively older studies.

Comment #6: Page 7, equation (1), why is no a concentration conversion factor $(1-\varphi)/\varphi$ in the reaction term? equation (2), I guess Db(z) is biodiffusion here, should describe it and also write down the function.

Response #6: This is because we express our reaction terms as [moles /volume sediment /time], and our concentrations as [moles/volume pore water] or [moles/volume solid phase]. In order to come to the right units in the total partial derivatives, one needs to correct only the concentrations (see e.g. Meysman et al., 2015).

$D\_b(z)$ is indeed the biodiffusion coefficient, which is assumed to be constant in a layer with thickness z_b (2 cm, after Dale et al. 2016 for the Palaeozoic) and had a value of 1 for the exponential coefficient $\lambda\_D\_b$ and 5 cm2 yr- for Db0 (after Dale et al. 2016 for the Palaeozoic) ;

$D\_b(z) = D\_b0$ for $z \leq z\_b$

$D\_b(z) = D\_b0$ exp-$(z-z\_b)/\lambda\_D\_b$ for $z > z\_b$

This form has been described before by Soetaert et al., 1996.

We will add a short description and an explicit mention of the function of the biodiffusion coefficient.

Dale A.W. et al. (2016) Geochim. Cosmochim. Acta. 189:251-268

Meysman F.J.R. et al. (2015) Geochim. Cosmochim. Acta. 152:122-141

Soetaert K. et al. (1996) Geochim. Cosmochim. Acta. 60 :1019-1040

Comment #7: Table 2. I don't know why SO4/(SO4+KSO4) is in in the reaction rate law of anaerobic oxidation of methane

Response #7: This is a typo and has to be corrected; the limitation of sulfate concentration is already included in the direct dependence of the AOM rate on the sulfate concentration. Thanks for noticing this.

Comment #8: Page 9, line 15. The sedimentary rate may have been changed across the P-T boundary.

Response #8: Yes, and we account for this as well in the timeseries simulation. However, as can be seen in Figure 5 and 6, the sedimentation rate has little effect on the sediment chemical zonation, and consequentially on the $\delta$13C variance.

Comment #9: Page 14, line 15. "The inverse relationship suggest that d13Ccarb variability is not controlled by the increased potential sample size." This doesn't make sense to me. If there is more sampling effort on short interval, it is possible to capture larger d13C variability. Could you show the relationship between the d13C variabilities and the numbers of data of different intervals?

Response #9: This has admittedly been written in an awkward manner and needs more clarification in an updated manuscript. Essentially the whole of section 3.1.3 (i.e. the subsampling routine) is already designed to mitigate this effect. Indeed sampling intensity increases over the P-Tr boundary beds and so we needed confirmation that this sampling artefact did not skew our results. This exercise is an integral part of the original work. The results of the subsampling routine are depicted in Figure 3. (Watercolour Regression curves), which show that sampling intensity does not create the pattern of increased $\delta$13C variability.

What has been written at line 15, merely points to the uneven distribution in time captured by some of the biozones. Units of longer duration could theoretically corroborate with more temporal variation of marine DIC- $\delta$13C. However, this does not seem to be the case, so we concluded that besides the changing sampling resolution, expanded biozones did not generate the $\delta$13C variability.

Comment #10: Figure 5 is good. Could you also show the influence of biodiffusion? The intensity of biodiffusion could be a function of oxygen level. It is better if bioirrigation is also included.

Response #10: We will include the effects of bio-diffusion and bio-irrigation on $\delta$13C variability in the updated manuscript. Hence, we will replace Figure 8 of the original work, and include the results of updated sensitivity tests (Figures 5 and 6 of the original work). To account for bio-irrigation as a non-local exchange process, in which pore water is exchanged with bottom water, we will introduce the following formulation:

I_irr(z) = $\alpha$(z) (C_ow - C(z))

The quantity $\alpha$(z) represents the depth-dependent irrigation intensity, and the solute concentrations of the bottom water and at depth are given by; C_ow and C(z), respectively. The attenuation of bio-irrigation intensity will be formulated as follows:

$\alpha$(z) = $\alpha$_0 exp(-z/X_irr)

where, $\alpha$_0 is the irrigation coefficient at the sediment water interface, and X_irr is the attenuation coefficient. For the baseline conditions we take Palaeozoic conditions for bio-diffusion and bio-irrigation that are based on Dale et al. 2016 (D_b0 = 5 cm2- yr-, z_b = 2 cm, $\alpha$_0 = 50 yr-, X_irr = 1 cm).

By conducting sensitivity experiments, we deduced the importance of changing these parameters under average OC sedimentation (730.5 $\mu$mol cm2- yr-, after van de Velde & Meysman 2016). Bio-irrigation attenuation with depth and the irrigation coefficient modulate the amplitude of the carbon isotope variations to some extent (Figure 1a of the response). However, there is effectively no difference in the effect on carbon isotope alteration trajectories between the post-extinction (burrowing organisms absent) and pre-extinction (burrowing organisms present). These parameters would only become important under present conditions with elevated sediment mixing (Figure 1 of the response).

In addition, we performed an updated sensitivity experiment for the remaining parameters of the sedimentary and marine environment (fraction of authigenic carbonate, sedimentation rate and marine oxygen, sulfate and DIC) under a changing OC flux (Figure 5 and 6 of the original work), and including the new parameters for bio-diffusion and bio-irrigation for the baseline model ($D\_b0 = 5$ cm2 yr-, $z\_b = 2$ cm, $\alpha\_0 = 50$ yr-, $X\_irr = 1$ cm). These tests do, however, not substantially deviate from the original work (Figure 2 of the response). So, we conclude that organic matter fluxes remain the dominant force behind bulk-rock carbon isotope alteration.

Van de Velde S. & Meysman, F. Aq. Geochem. 22: 469-504

Dale A.W. et al. (2016) Geochim. Cosmochim. Acta. 189:251-268

Comment #11: Page 22: The discussions about the influence of seawater chemistry and meteoric diagenesis are great. It may be better to talk about the other explanations for the scatter d13C such as heterogeneity in seawater chemistry, meteoric diagenesis and the variation of mineralogy in one section. Also, all the other discussions can be putted in another section. These may make the structure of the discussion part more clear.

Response #11: Agreed. The discussion will be restructured.
* * *
[Figure]

**Fig. 1.** Sensitivity experiments for bio-irrigation and bio-diffusion on the $\Delta 13C_{primary-bulk}$ of diagenetic altered carbonate under a normal OC flux (730.5 $\mu$mol cm2-).

[Figure]

**Fig. 2.** Sensitivity experiments for a range of parameters of the sedimentary and marine environment on diagenetic altered carbonate (Palaeozoic bio-irrigation and bio-diffusion included).

---

## Author Comment (AC2) · 31 Jul 2017

On behalf of my co-authors, I would like to thank the reviewer for the detailed and constructive assessment of our manuscript. We have thoroughly considered the issues raised in the reviews of the original manuscript, and we provide a point-by-point response below.

Comment #1: I have two primary concerns for this paper, both related to the treatment of carbonate chemistry. First, how does the diagenetic (reactive-transport) model account for changes in the carbonate system? It seems that the model considers CO aqueous, bicarbonate, and DIC, but this is incomplete. Bicarbonate/CO2 additions and subtractions to the DIC pool do not simply add/subtract to those supplies. Additions of those species will cause re-equilibrium among all species. These re-equilibrations will results in non-linear changes to calcite saturation.

Response #1: The carbonate chemistry is not accounted for in the current model, instead we chose to represent calcification by equation (4), the same approach used in other studies such as those of Fantle and DePaolo. We did so, for one important reason: although carbonate saturation is an important parameter that can be modulated by microbial activity, the extracellular polymeric substance produced by the same microbes has been invoked as well to control calcite nucleation. Considering this uncertainty we preferred a rate expression (equation 4) that does not put one particular mode of carbonate production over the other. So, in extension, there is also no re-equilibration among DIC species envisioned in the current model. In addition, it is quite challenging to incorporate parameters, such as pH, in these types of diagenetic models.

In our opinion, a more in depth understanding of microbial-steered calcification would be needed before carbonate chemistry could be included in models, similar to introduced here.

Fantle M.S. & DePaolo D.J. (2006) Geochim. Cosmochim. Acta. 70 : 3883-3904

Fantle M.S. & DePaolo D.J. (2007) Geochim. Cosmochim. Acta. 71 : 2524-2546

Comment #2: On a related note, I don't understand the kinetic rate expressions that are provided, and the explanations are insufficient. The authors cite Boudreau 1997 checked the reference and could not find an explanation, though it could be because the reference is a ∼400pg book. Why are all of the rate expression factors of O2 and SO42-? How does a monod constant compare to a solubility product or rate constant? Obviously I am not as well versed in these things as the authors of this paper, but I

asked to other scientists in related fields and they were similarly confused. The authors must make these concepts more accessible to their target audience. Citing a 400 pg book is not sufficient.

Response #2: These rate expressions are the classical equations, as they have been used for the past 20 years (e.g. Meysman et al., 2003; Soetaert et al., 1996; Berg et al., 2003; Van Cappellen and Wang, 1996). Monod constants are factors that represent the inhabitation/limitation of a certain electron acceptor (in this case $O_2$ or $SO_4^{2-}$). At high concentrations of oxygen, oxygen reduction will be efficient and other mineralisation pathways will be inhibited. At lower concentrations, oxygen reduction will be less efficient, and other pathways will gain importance. High sulfate concentrations will then in its turn inhibit methanogenesis. This leads to the chemical zonation as described by Froelich et al. (1979).

Although we appreciate that not everybody is experienced with the literature on theoretical modelling of sediment diagenesis, we do not think a detailed description of these formulations is required, as these have been standard for decades. We will change the text in order to make the meaning of a monod constant versus a rate constant and solubility constant more clear and we do understand that the reference might be not appropriate for people who want to find detailed descriptions, and therefore we will update these with the once mentioned above.

Meysman F.J.R. et al. (2003) Comp. Geosci. 29:301-318

Soetaert K. et al. (1996) Geochim. Cosmochim. Acta. 60 :1019-1040

Van Cappellen and Wang (1996) Am. J. Sci. 296:197-243

Berg et al. (2003) Am. J. Sci. 303:905-955

Froelich P.N. et al. (1979) Geochim. Cosmochim. Acta. 43:1075-1090

Comment #3: Second the authors assume constant boundary conditions for their model, but also recognize changes in the carbon isotope composition of seawater DIC

as a result of "changes in the sources and sinks of the long term (>100 ky) carbon cycle." The changes in the carbon cycle are also affecting changes in the carbonate chemistry, which should affect the ability of authigenic carbonate to precipitate and how much dissolution/equilibration takes place. See Payne et al., 2010 PNAS to start.

Response #3: We thank the reviewer for this important suggestion. Although carbonate chemistry is not explicitly incorporated in the model, the observations, as made in the Payne et al., (2010), could be envisioned to have consequences for the adopted DIC concentration and carbonate recrystallization/ dissolution rate (equation 4). We performed a new sensitivity test, and judged that increased DIC concentration (2–7 $\mu$m cm-3 DIC, Ridgwell 2005) can dampen $\delta$13C variability, but only to a very small degree, see Figure 1d (of the response) below.

Ridgwell A. (2005) Mar. Geol. 217: 339-357

According to Payne et al, transient ocean acidification at the end-Permian mass extinction would have stimulated carbonate dissolution. If the DIC pool increased over the studied time interval, this change would have suppressed the temporal trend in the amplitude of the $\delta$13C variance, as observed for both Iran and China. In addition, dissolved carbonate, but also alkalinity input via weathering, could have increased the post-extinction carbonate ion inventory. Carbonate saturation could, in such scenario, elevate the production of authigenic carbonates. When accepting both observations, this would in addition require a modulation of our simulated carbonate reactivity (i.e., expressed in equation 4) over the Permian–Triassic interval. As reactivity of carbonate equates to more $\delta$13C variability, this would largely cancel out concomitant suppression of $\delta$13C variability by an enlargement of the DIC pool (see above Figure 1d). As such these two parameters would largely oppose each other. Since, we do not know how to scale the carbonate reactivity (justifiable to the natural situation); we think that it is better to dismiss temporal variation in this parameter, at least, in the current model simulation. A model that incorporates all the nuances associated with biologically-controlled (or microbial-induced) calcite precipitation, dissolution and overall carbonate

chemistry would presumably be an answer to the lack of this control. But, as already outlined, we think that far too many variables would be loosely (or un)constrained in such model, with our current knowledge of these processes. Hence, validation of our simple model remains more trustworthy and straightforward to interpret.

However, we will change the boundary conditions for DIC, from 2.2 $\mu$m cm-3 (modern) to 4.5 $\mu$m cm-3 (Payne et al. 2010), to better represent the Permian situation. These adjustments will slightly suppress the generated $\delta$13C variability (see above Figure 1d). Hence, we will replace Figure 8 of the original work, and include the results of the updated sensitivity test (Figures 5 and 6).

Comment #4: Below are some minor comments I have on the text and figures: The authors often use the term "sedimentation rate", which I take to mean F_carb? They do not seem to include F_OC under this definition? Regardless, the use of "sedimentation rate" is confusing and should be made more specific. Perhaps "carbonate sedimentation rate" or "calcite sedimentation rate".

Response #4: This will be implemented where applicable.

Comment #5: Pg. 3, line 22 : What do the authors mean by "High carbonate ion concentrations"? Are we talking about DIC or CO32-? If these high values are predicted, why did the authors use modern DIC values in their model?

Response #5: This statement was not very precise and we will refer to the larger DIC pool, as discussed in Payne et al. (2010) and Ridgwell (2005), for the Permian–Triassic ocean. In addition, the boundary conditions of the model will be changed (as discussed above).

Comment #6: Pg. 3, line 26: "High carbonate ion concentrations are invoked.." True, and this implies a change in seawater carbonate chemistry that should be considered in the time series diagenetic model.

Response #6: This change will be implemented.

Comment #7: Pg. 4, lines 7–15: The phrases in italics are confusing. I am having a hard time parsing their meaning and tying them to clauses before them. Are they even used later? I would just get rid of them.

Response #7: These phrases return in the section dealing with time series simulation (Section 5). But we will omit these phrases in the revised version.

Comment #8: Pg.4, lines 16: should be studied". To carry out this investigation" implies action in the past.

Response #8: Ok

Comment #9: Pg. 6, lines 3–13: Can the authors reference a figure here? It would be easier to follow the explanation with a visual aid.

Response #9: Ok

Comment #10: Section 3.2.1: Have the authors considered the role of other metabolisms, such as Fe and Mn reduction? These metabolisms can yield much more alkalinity than sulphate reduction (see Bergmann et al., 2013 in Palaios). I suspect that Fe reduction is not quantitatively important for most of the Phanerozoic when sulphate and O2 are high, but if sulphate and O2 is low at the P/T it could be significant.

Response 10: Alkalinity and carbonate chemistry do not play a role in the current model construction (see above). The selection was primarily based on whether carbonate precipitates formed by these specific metabolic pathways have been described in the literature. In addition, the quantitative importance of sedimentary OC consumption by aerobic heterotrophs, sulfate reducing microbes and methanogens was used as a selection criterion.

Comment #11: Pg. 10, lines 10–12: "The previous…solid carbonate." This sentence is hard to follow. Please consider clarifying.

Response #11: Ok

Comment #12: Figure 1: The median line is completely obscured for the Iran plot. Can the authors move the line to in front of the data points?

Response #12: Ok

Comment #13: Pg. 14, line 27: "depleted and "heavier" are incorrect terms. Should be "lower" and "higher". From Sharp 2007 (Principle of Stable Isotope Geochemistry pg. 16): "As numbers, delta values can be high or low, positive or negative, but not heavy or light,(nor can they be). . .depleted or enriched."

Response #13: This will be corrected accordingly.

Comment #14: Figure 3: What is the range of values represented by the green colors in the biozone thickness/duration graphs? Does more saturated mean shorter or longer?

Response #14: This will be clarified.

Comment #15: Figure 7: I don't understand the point of this figure. I understand that the authors are changing the distributions from which the F_OC is generated in the model, but the rest of figure is lost on me.

Response #15: This will be better explained in the revised version.

Comment #16: Pg. 19, line 5: Can the authors elaborate in how D13C_primar-bulk was derived in Schobben et al, 2016? It would be usefull to do here, so the reader can understand the paper without first reading the other.

Response #16: A full explanation will be given in the revised version.

––––––––––––––––––––––––––

[Figure]

**Fig. 1.** Sensitivity experiments for a range of parameters of the sedimentary and marine environment on diagenetic altered carbonate (Palaeozoic bio-irrigation and bio-diffusion included, see AC1)

---

## Author Response (AR1)

**School of Earth and Environment**

University of Leeds
Leeds LS2 9JT

m.schobben@leeds.ac.uk

[Figure]

Dear Dr Winguth,

On behalf of my co-authors, I would like to thank you and the reviewers for the detailed and constructive comments. We have thoroughly considered the issues raised in the reviews. Hereby, we submit a response letter to these comments, an annotated version of the revised manuscript (including a list of all changes at the end of the document) as well as clean version of the revised manuscript.

Besides improvements to the diagenetic model, this new version includes a more in-depth discussion on limestone lithification, and how these post-depositional processes might have affected the chemical composition of carbonate rock. The discussion now also includes more reference to diagenetic resetting of carbonate- C isotope composition by the interaction with meteoric fluids. We deem it, however, of high importance to study diagenetic mechanisms that do not occur by the interaction with meteoric fluids. The introduced diagenetic model is capable of explaining the total range as well as the temporal patterns of residual C isotope variability, superimposed on the long-term negative C isotope trend straddling the Permian–Triassic boundary. This study thereby provides an alternative model for diagenetic C isotope resetting, which warrants consideration.

We regard this study to be of importance as it has general implications for C isotope-based chemostratigraphic studies, by evaluating limitations in its use, as well as for studies that link carbonate C isotope composition to the biogeochemical carbon cycle.

Please find a detailed response to all comments on the accompanying pages of this letter.

Sincerely,

Martin Schobben

**Comments by the editor (A. Winguth)**

I share concerns with one reviewer, that delta-13C records may have been also generated by meteoric diagenesis.

Response: The assertion that exposure of carbonate sediments to meteoric water is the predominant mechanism of limestone lithification is a long-standing paradigm. Moreover, and perhaps as a result of this view, chemical alteration of carbonate rock (i.e., resetting of carbonate C and O isotope signals) is often assigned to this specific diagenetic environment. However, there are a number of caveats in applying this model uniformly to the whole of the carbonate archive in general, and the here-studied rock sequences specifically.

Firstly, studies on carbonate rock lithification are heavily biased towards Pleistocene carbonate platforms, which have been inundated by meteoric fluids during global sea-level lowstands associated with glacials. Although opposing views exist, there is some consensus that the Permian–Triassic boundary interval was a greenhouse world. Therefore these studies on (relatively) young carbonate deposits might be of little relevance to these ancient rock units. On top of that, the general trend was that of a sealevel rise for the studied time interval. The lack of supporting evidence for subaerial exposure shortly after deposition of the carbonate sediment (the time window during which the sediments are most reactive and are highly porous) requires alternative modes for diagenetic carbonate rock stabilization. Textural and chemical evidence diagnostic of diagenesis by meteoric water is generally absent, based on petrological studies. However, one study suggests partial recrystallization of the Meishan sequence invoked by interaction with meteoric fluids (Li & Jones, 2017). These authors cite textural features (zoned dolomite crystals) as diagnostic markers for meteoric water-controlled diagenetic alteration. However, many studies have shown that such supposed diagnostic markers (textural and chemical) are ambiguous, and can also be the results of burial and marine diagenesis (e.g., Melim et al., 2002).

So, we cannot completely discount the influence of meteoric water on the isotope signatures. The previous arguments do, however, show the importance of evaluating the influence of alternative modes of diagenesis on carbonate $\delta^{13}C$. Our conceptual model is an exercise to show that diagenetic pathways, fuelled by marine organic matter, can produce the observed patterns. This does not necessarily mean that the real variation is—in all instances—caused by this mechanism, but we merely point to its excellent potential to explain the magnitude of the variations and the temporal pattern of variability across the P–Tr boundary

Action: In order to better bring across these points in the revised manuscript, we added a more extended reference to the ideas above in the introduction and added a more in-depth discussion concerning evidence for meteoric fluid infiltration and subsequent diagenetic resetting of isotope signals on the studied suites of rock.

Melim et al. (2002) *Marine Geology* 185: 27–53
Li & Jones (2017) *Palaeogeogr. Palaeoclimatol. Palaeoecol.* 468: 18–33

Another major concern of the other reviewer is a more detailed description of the diagenetic (reactive transport) model in particular how changes in the carbonate system are considered.

Response: We agree that some of the aspects of the diagenetic model were underdeveloped or missing in the original work. We refer to the reviewer's specific comments, where we delineate our course-of-action concerning these specific issues.

**Comments by anonymous reviewer #1**

The authors try to interpret the scatter in the d13C records as recrystallization with an organic carbon source during marine diagenesis. However, it is also possible that some of these variations may have been generated by meteoric diagenesis. Thus, more discussions on this point are required, and it's better to include some petrography or geochemical evidences. The variations may also be generated by changes of rock type, minerals and calcified fossil species. Thus more descriptions on the samples are necessary.

Response: We fully acknowledge the influence of lithology, polymorphism and polymorph-specific C isotope values on variations superimposed on the first-order trend as well as the potential importance of meteoric water in diagenetic C isotope alteration. However, we like to stringently point out that the conceptualized model is merely an exercise to show the potential of diagenetic pathways fuelled by marine organic matter in inducing the here observed patterns. This does not necessarily mean that the real variation is—in all instances—caused by this mechanism, but we point to its excellent potential to explain the magnitude of the variations as well as simulate the temporal pattern of variability across the P–Tr boundary.

Action: We extended our discussion (section 6.1) to include a more in-depth reference to; how meteoric –controlled carbonate stabilization (i.e., cementation) is thought to occur, and refer to studies that looked for diagnostic petrological evidence of meteoric diagenesis in the studied rock sequences. Furthermore, we included more text in the introduction (section 1.1), specifying the, perhaps, somewhat skewed views that prevail in studies concerning limestone diagenesis, and point to the need to study possible alternatives. We, however refrain from incorporating a petrological study, as it would substantially increase the length of the work, and might still not give a 100% accurate understanding of carbonate diagenetic trajectories (in reference again to the ambiguity of many of the alleged diagnostic markers, e.g. , Melim et al., 2002).

Melim et al. (2002) *Marine Geology* 185: 27–53

The data compiled from the Meishan section in South China maybe can preclude the influence of seawater chemistry. However, the Iran data do not come from the same site. The scatter in d13C may be generated by spatial heterogeneity in seawater chemistry rather than diagenesis.

Response: We fully agree with this. However, trends in residual $\delta^{13}C$ variability recorded in both regions reproduce the same temporal pattern (Figure 2). This pattern cannot easily be explained by local processes (heterogeneity of water masses), and rather points to the existence of an overarching mechanism (OC-steered diagenesis). This alterative gives the most parsimonious explanation, incorporating evidence for increased heterogeneity of OC accumulation and authigenic mineral additions.

Action: Additional discussion has been added in the revised work, in order to clarify this nuance (section 6.2).

The authors have also talked about authigenic carbonates. A definition on authigenic carbonates is required. If the carbonates were mainly formed by recrystallization, their d13C value may have been changed due to the exchange of carbonate ions with pore-water. Can this type of carbonates be classified into authigenic carbonates? Except some shells, all ancient carbonates have suffered from recrystallization. Does this means that all the ancient carbonates are authigenic carbonates?

Response: Mixing of dissolved carbonate and metabolic derived carbon is fully accounted for in the current model by a recrystallization process. This is visually expressed in Figure 4b,d and mathematically expressed as equation (4), the latter expression accounts for the rate of dissolution and crystallization. Since dissolution adds new DIC to the porewaters, this constant exchange has a somewhat buffering effect on the addition of metabolic derived carbon.

We do not infer that all ancient carbonates are authigenic, and we agree with the reviewer that our definitions are rather imprecise. In the original manuscript we lump; ongoing equilibrium recrystallization with depth together with microbial-mediated carbonate cementation (at the sediment water interface), both under the label of authigenic carbonates. The results of recrystallization are now termed diagenetically altered carbonate rock. In this definition, diagenetically altered carbonate has partially dissolved and replaced by less reactive components. Whereas authigenic carbonate is formed at the sediment–water interface.

Action: We will include an updated definition in the revised manuscript, where we divide between diagenetic carbonate alteration and authigenic carbonate addition (section 3.2.3).

Page 2, line 13 methanogenesis belongs to degradation of organic matter.

Action: Methanogenesis will be omitted.

Page 5, section 3.1.2. Some of the d13C data for the Meishan section are very old and are less -5‰ which may be generated by analytical error.

Response: The precision of carbonate carbon isotope measurements has not changed over past decades, and therefore does not introduce an increasing measurement bias with progressively older studies.

Page 7, equation (1), why is no a concentration conversion factor $(1-\varphi)/\varphi$ in the reaction term? equation (2), I guess Db(z) is biodiffusion here, should describe it and also write down the function

Response: This is because we express our reaction terms as [moles /volume sediment /time], and our concentrations as [moles/volume pore water] or [moles/volume solid phase]. In order to come to the right units in the total partial derivatives, one needs to correct only the concentrations (see e.g. Meysman et al., 2015).

$D_b(z)$ is indeed the biodiffusion coefficient, which is assumed to be constant in a layer with thickness $z_b$ (2 cm, after Dale et al. 2016 for the Palaeozoic) and had a value of 1 for the exponential coefficient $\lambda_{Db}$ and 5 $cm^2$ $yr^{-1}$ for $D_{b0}$ (after Dale et al. 2016 for the Palaeozoic) ;

$$D_b(z) = D_{b0}$$
$$D_b(z) = D_{b0}.e^{-(z-z_b)/\lambda_{Db}}$$

for

$$z \leq z_b$$
$$z > z_b$$

This form has been described before by Soetaert et al., 1996.

Dale A.W. et al. (2016) *Geochim. Cosmochim. Acta.* 189:251-268
Meysman F.J.R. et al. (2015) *Geochim. Cosmochim. Acta.* 152:122-141
Soetaert K. et al. (1996) *Geochim. Cosmochim. Acta. 60* :1019-1040

Action: We added a short description and an explicit mention of the function of bio-diffusion in section 3.2.1

Table 2. I don't know why SO4/(SO4+KSO4) is in in the reaction rate law of anaerobic oxidation of methane

Response:  We thank for noticing this issue. This is a typo and is now corrected; the limitation of sulfate concentration is already included in the direct dependence of the AOM rate on the sulfate concentration.

Page 9, line 15. The sedimentary rate may have been changed across the P-T boundary.

Response: We thank for the insightful comment. After restructuring of the model code we found a modest change induced by the sedimentation rate.

Action: We now force the time series simulation with a change in sedimentation concomitant with the extinction boundary; following suggestions by multiple studies that the sedimentation rate increased after this event (Algeo et al., 2010). However, this change does not significantly alter the pattern of increasing $\delta^{13}$C variability across the P–Tr transition.

Algeo and Twitchett (2010) G*eology* 38: 1023–1026

Page 14, line 15. "The inverse relationship suggest that d13Ccarb variability is not controlled by the increased potential sample size." This doesn't make sense to me. If there is more sampling effort on short interval, it is possible to capture larger d13C variability. Could you show the relationship between the d13C variabilities and the numbers of data of different intervals?

Response: This has admittedly been written in an awkward manner. Essentially the whole of section 3.1.3 (i.e. the subsampling routine) is already designed to mitigate this effect. Indeed sampling intensity increases over the P–Tr boundary beds and so we needed confirmation that this sampling artifact did not skew our results.  This exercise is an integral part of the original work. The results of the subsampling routine are depicted in Figure 2 (Watercolor Regression curves), which shows that sampling intensity does not create the pattern of increased $\delta^{13}$C variability.

What has been written at line 15 merely points to the uneven distribution in time captured by some of the biozones. These temporally more expanded units could theoretically corroborate

with more temporal variation of marine DIC- $\delta^{13}$C. However, this does not seem to be the case, so we concluded that besides the changing sampling resolution, temporally more expanded biozones did not generate the $\delta^{13}$C variability.  So, we conclude that the variability relates to a real (i.e. diagenetic) signal.

Action: Some changes have been made in the respective text fragment of section 4.2. We deem it unnecessary to include the number of data points, as the constructed subsampling routine already abolishes the effect of differential data density on the observed temporal trends. This issue seems largely related to an imperfect description of this biozone length-related issue in the respective text fragment, which has been fixed now.

Figure 5 is good. Could you also show the influence of biodiffusion? The intensity of biodiffusion could be a function of oxygen level. It is better if bioirrigation is also included.

Response: We included the effects of bio-diffusion and bio-irrigation on $\delta^{13}$C variability in the updated manuscript.  Hence, we updated Figure 8 of the original work, and included the results of updated sensitivity tests (Figures 5 and 6). To account for bio-irrigation as a non-local exchange process, in which pore water parcels are exchanged with bottom water parcels, we introduced the following formulation:

$$I_{irr}(z) = \alpha(z)(C_{ow} - C(z))$$

The quantity $\alpha(z)$ represents the depth-dependent irrigation intensity, and the solute concentrations of the bottom water and at depth are given by; $C_{ow}$ and $C(z)$, respectively. The attenuation of bio-irrigation intensity is formulated as follows:

$$\alpha(z) = \alpha_0 \exp(-\frac{z}{x_{irr}})$$

where $\alpha_0$ is the irrigation coefficient at the sediment water interface, and $x_{irr}$ is the attenuation coefficient. For the baseline conditions we took Paleozoic conditions for bio-diffusion and bio-irrigation that are based on Dale et al. 2016 ($D_{b0}$ = 5 cm$^{2-}$ yr$^-$, $z_b$ = 2 cm, $\alpha_0$ = 50 yr$^-$, $x_{irr}$ = 1 cm).

By conducting sensitivity experiments, we deduced the importance of changing these parameters under average OC sedimentation (730.5 µmol cm$^{2-}$ yr$^-$, after van de Velde & Meysman 2016). Bio-irrigation attenuation with depth and the irrigation coefficient modulate the amplitude of the carbon isotope variations to some extent (Figure 5 and 6). However, there is effectively no difference in the effect on carbon isotope alteration trajectories between the post-extinction (burrowing organisms absent) and pre-extinction (burrowing organisms present). These parameters would only become important under present conditions with elevated sediment mixing (Figure 5 and 6).  **So, we conclude that organic matter fluxes remain the dominant force behind bulk-rock carbon isotope alteration.**

van de Velde, S.  & Meysman, P. (2016) *Aquatic Geochemistry* 22: 469–504

Page 22: The discussions about the influence of seawater chemistry and meteoric diagenesis are great. It may be better to talk about the other explanations for the scatter d13C such as heterogeneity in seawater chemistry, meteoric diagenesis and the variation of

mineralogy in one section. Also, all the other discussions can be putted in another section. These may make the structure of the discussion part more clear.

Action: We restructured the discussion intro three separate sections;

1) Section 6.1 *Stochastic carbon isotope signatures of the Permian–Triassic boundary beds*: dealing with meteoric diagenesis on the here-studied rock sequences and changes in polymorph assemblage and their polymorph-specific isotope signals.

2) Section 6.2 *The carbon isotopic composition of the Permian--Triassic DIC reservoir:* solely discussing actual changes in water column DIC isotopic changes (local vs global) and how they can be read from the bulk carbonate C isotope record.

3) Section 6.3 *Perspectives on the Permian–Triassic carbonate archive: stratigraphy, biological evolution and the global biogeochemical carbon cycle*:  dealing with the major implications of the here-introduced model of diagenetic carbonate alteration.

**Comments by anonymous reviewer #2**

I have two primary concerns for this paper, both related to the treatment of carbonate chemistry. First, how does the diagenetic (reactive-transport) model account for changes in the carbonate system? It seems that the model considers CO aqueous, bicarbonate, and DIC, but this is incomplete. Bicarbonate/$CO_2$ additions and subtractions to the DIC pool do not simply add/subtract to those supplies. Additions of those species will cause re-equilibrium among all species. These re-equilibrations will results in non-linear changes to calcite saturation.

Response: The carbonate chemistry is not accounted for in the current model, instead we chose to represent calcification by equation (7), similar to the approach in the studies by Fantle and Depaolo.  We did so, as alternative drivers of calcification (besides carbonate chemistry) have been postulated; e.g. the extracellular polymer substances produced by the same microbes have as well been invoked to control calcite nucleation (Arp and Reitner , 2001). Considering this uncertainty we preferred a rate expression (equation 7) that does not put one particular mode of carbonate production over the other. So, in extension, there is also no re-equilibration among DIC species envisioned in the current model. In addition, it is quite challenging to incorporate parameters, such as pH, in these types of diagenetic models.

In our opinion, a more in depth understanding of microbial-steered calcification would be needed before carbonate chemistry could be included in models like ours.

Arp G. & Reitner R.J. (2001) *Science* 292: 1701-1704
Fantle M.S. & DePaolo D.J. (2006) *Geochim. Cosmochim. Acta.* 70 : 3883-3904
Fantle M.S. & DePaolo D.J. (2007) *Geochim. Cosmochim. Acta.* 71 : 2524-2546

Action: To further clarify that our model does not include carbonate chemistry, we added additional specific mentions to our definition of carbonate diagenesis, and why we deem our formulation the most appropriate solution (section 3.2.3).

On a related note, I don't understand the kinetic rate expressions that are provided, and the explanations are insufficient. The authors cite Boudreau 1997 checked the reference and could not find an explanation, though it could be because the reference is a ~400pg book. Why are all of the rate expression factors of O2 and SO42-? How does a monod constant compare to a solubility product or rate constant? Obviously I am not as well versed in these things as the authors of this paper, but I asked to other scientists in related fields and they were similarly confused. The authors must make these concepts more accessible to their target audience. Citing a 400 pg book is not sufficient.

These rate expressions are the classical equations, widely used for the past 20 years (e.g. Meysman et al., 2003; Soetaert et al., 1996; Berg et al., 2003; Van Cappellen and Wang, 1996). Monod constants are factors that represent the inhabitation/limitation of a certain electron acceptor (in this case O2 or SO42-). At high concentrations of oxygen, oxygen reduction will be efficient and other mineralization pathways will be inhibited. At lower concentrations, oxygen reduction will be less efficient, and other pathways will gain importance. High sulfate concentrations will then in its turn inhibit methanogenesis. This leads to the chemical zonation as described by Froelich et al. (1979).

Meysman F.J.R. et al. (2003) Comp. Geosci. 29:301-318
Soetaert K. et al. (1996) Geochim. Cosmochim. Acta. 60 :1019-1040
Van Cappellen and Wang (1996) Am. J. Sci. 296:197-243
Berg et al. (2003) Am. J. Sci. 303:905-955
Froelich P.N. et al. (1979) Geochim. Cosmochim. Acta. 43:1075-1090

Action: We changed the text (section 3.2.1) in order to make the meaning of a Monod constant versus a rate constant and solubility constant more clear, as we do understand that the reference might not be appropriate for all readers.

Second the authors assume constant boundary conditions for their model, but also recognize changes in the carbon isotope composition of seawater DIC as a result of "changes in the sources and sinks of the long term (>100 ky) carbon cycle." The changes in the carbon cycle are also affecting changes in the carbonate chemistry, which should affect the ability of authigenic carbonate to precipitate and how much dissolution/equilibration takes place. See Payne et al., 2010 PNAS to start.

Response: We thank the reviewer for this important suggestion. Although carbonate chemistry is not explicitly incorporated in the model, the observations as made in the Payne et al., 2010 PNAS paper could be envisioned to have consequences for the adopted DIC concentration and carbonate recrystallization/dissolution rate (equation 7).

According to Payne et al, transient ocean acidification at the EPME would have stimulated carbonate dissolution. If the DIC pool increased over the studied time interval, this change would have suppressed the temporal trend in the amplitude of the $\delta^{13}C$ variance, as observed for both Iran and China.  In addition, dissolved carbonate (and also alkalinity input via weathering), could have increased the post-extinction carbonate ion inventory. Carbonate saturation could, in such scenario, elevate the production of authigenic carbonates. Accepting both observations would require a modulation of our simulated carbonate reactivity (i.e., expressed as equation 7) over the Permian–Triassic interval. As reactivity of carbonate

equates to more $\delta^{13}$C variability, this would largely cancel out concomitant suppression of $\delta^{13}$C variability by an enlargement of the DIC pool. As such these two parameters would largely oppose each other. Since, we do not know how to scale the carbonate reactivity (justifiable to the natural situation); we think that it is better to dismiss temporal variation in this parameter, at least, in the current model simulation. A model that incorporates all the nuances associated with biologically controlled (or microbial-induced) calcite precipitation, dissolution and overall carbonate chemistry would presumably be an answer to the lack of this control. But, as already outlined, we think that too many variables would be loosely (or un)constrained in such model, with our current knowledge of these processes. Hence, validation of our simple model remains more trustworthy and straightforward to interpret.

Action: We performed a new sensitivity test, and judged that increased DIC concentration (2–7 $\mu$m cm$^{-3}$ DIC, Ridgwell 2005) can dampen $\delta^{13}$C variability, but only to a very small degree. In addition, we changed the boundary conditions for DIC, from 2.2 $\mu$m cm$^{-3}$ (modern) to 4.5 $\mu$m cm$^{-3}$ (Payne et al. 2010 PNAS) to better represent the Permian situation. These adjustments slightly suppress the generated $\delta^{13}$C variability. Hence, we replaced Figure 8 of the original work, and included the results of the updated sensitivity test (Figures 5 and 6).

Ridgwell A. (2005) *Mar. Geol.* 217: 339-357

Below are some minor comments I have on the text and figures: The authors often use the term "sedimentation rate", which I take to mean F_carb? They do not seem to include F_OC under this definition? Regardless, the use of "sedimentation rate" is confusing and should be made more specific. Perhaps "carbonate sedimentation rate" or "calcite sedimentation rate".

We have further clarified this. When we refer to sedimentation rate of the combined solids and solutes, we added the symbols: *w* and *v*. This addition makes it possible to better distinguish "sedimentation rate" from instances where we refer to the flux of OC.

Pg. 3, line 22 : What do the authors mean by "High carbonate ion concentrations"? Are we talking about DIC or CO32-? If these high values are predicted, why did the authors use modern DIC values in their model?

This statement was not very precise and we now refer to the larger DIC pool, as discussed in Payne et al. 2010 PNAS and Ridgwell 2005 Marine Geology, for the Permian–Triassic ocean. In addition, the boundary conditions of the model have been changed (see discussion above).

Pg. 3, line 26: "High carbonate ion concentrations are invoked.." True, and this implies a change in seawater carbonate chemistry that should be considered in the time series diagenetic model.

This change has been implemented.

Pg. 4, lines 7–15: The phrases in italics are confusing. I am having a hard time parsing their meaning and tying them to clauses before them. Are they even used later? I would just get rid of them.

We omitted these phrases in the revised version.

Pg.4, lines 16: should be studied". To carry out this investigation" implies action in the past.

OK

Pg. 6, lines 3–13: Can the authors reference a figure here? It would be easier to follow the explanation with a visual aid.

Ok, as a consequence the Figures 1 and 3 (of the original) have now moved to the front, and the numbering has changed (Fig. 1 and 2 in the revised version).

Section 3.2.1: Have the authors considered the role of other metabolisms, such as Fe and Mn reduction? These metabolisms can yield much more alkalinity than sulphate reduction (see Bergmann et al., 2013 in Palaios). I suspect that Fe reduction is not quantitatively important for most of the Phanerozoic when sulphate and $O_2$ are high, but if sulphate and $O_2$ is low at the P/T it could be significant.

Alkalinity and carbonate chemistry do not play a role in the current model construction (see above). The selection was primarily based on whether carbonate precipitates formed by these specific metabolic pathways have been described in the literature. In addition, the quantitative importance of sedimentary OC consumption by aerobic heterotrophs, sulfate reducing microbes and methanogens was used as a selection criterion.

Pg. 10, lines 10–12: "The previous…solid carbonate." This sentence is hard to follow. Please consider clarifying.

OK

Figure 1: The median line is completely obscured for the Iran plot. Can the authors move the line to in front of the data points?

OK

Pg. 14, line 27: "depleted and "heavier" are incorrect terms. Should be "lower" and "higher". From Sharp 2007 (Principle of Stable Isotope Geochemistry pg. 16): "As numbers, delta values can be high or low, positive or negative, but not heavy or light,(nor can they be)…depleted or enriched."

This has been corrected accordingly.

Figure 3: What is the range of values represented by the green colors in the biozone thickness/duration graphs? Does more saturated mean shorter or longer?

This has been clarified.

Figure 7: I don't understand the point of this figure. I understand that the authors are changing the distributions from which the F_OC is generated in the model, but the rest of figure is lost on me.

The figure has been simplified and now only refers to how the spatial OC variability and its consequences for spatial C isotope variability.

Pg. 19, line 5: Can the authors elaborate in how D13C_primar-bulk was derived in Schobben et al, 2016? It would be useful to do here, so the reader can understand the paper without first reading the other.

A full explanation has been given in the revised version.

[revised manuscript text omitted]
). M̶T̶h̶e̶ ̶e̶q̶u̶a̶t̶i̶o̶n̶s̶ ̶t̶o̶ ̶d̶e̶s̶c̶r̶i̶b̶e̶ ̶t̶h̶e̶ ̶r̶e̶a̶c̶t̶i̶v̶e̶-̶t̶r̶a̶n̶s̶p̶o̶r̶t̶ ̶m̶o̶d̶e̶l̶ ̶h̶a̶v̶e̶ ̶t̶w̶o̶ ̶t̶r̶a̶n̶s̶p̶o̶r̶t̶ ̶p̶a̶r̶a̶m̶e̶t̶e̶r̶s̶:̶ ̶m̶o̶l̶e̶c̶u̶l̶a̶r̶ ̶d̶i̶f̶f̶u̶s̶i̶o̶n̶ ̶o̶f̶ ̶s̶o̶l̶u̶t̶e̶s̶ ̶($D_i$)̶ ̶a̶n̶d̶ ̶s̶e̶d̶i̶m̶e̶n̶t̶a̶t̶i̶o̶n̶ ̶r̶e̶p̶r̶e̶s̶e̶n̶t̶e̶d̶ ̶b̶y̶ ̶d̶o̶w̶n̶w̶a̶r̶d̶ ̶a̶d̶v̶e̶c̶t̶i̶o̶n̶ ̶o̶f̶ ̶t̶h̶e̶ ̶s̶o̶l̶i̶d̶ ̶a̶n̶d̶ ̶s̶o̶l̶u̶t̶e̶.̶ ̶M̶olecular diffusion for porewater solutes is expressed by Fick's first law, where $D_i$ (the effective diffusion coefficient) is calculated following the definition as given in Boudreau and Meysman (2006).

$$D_i = \frac{D_0}{(1 - 2\ln\varphi)} \tag{3}$$

20   Where $D_0$ is a function of temperature and salinity and has been calculated with the R package *marelac* (Soetaert et al., 2016) ̶a̶n̶d̶ ̶c̶o̶r̶r̶e̶c̶t̶e̶d̶ ̶f̶o̶r̶ ̶t̶o̶r̶t̶u̶o̶s̶i̶t̶y̶,̶ ̶f̶o̶l̶l̶o̶w̶i̶n̶g̶ ̶e̶q̶u̶a̶t̶i̶o̶n̶ ̶3̶,̶ ̶d̶e̶r̶i̶v̶i̶n̶g̶ ̶t̶h̶e̶ ̶e̶f̶f̶e̶c̶t̶i̶v̶e̶ ̶d̶i̶f̶f̶u̶s̶i̶o̶n̶ ̶c̶o̶e̶f̶f̶i̶c̶i̶e̶n̶t̶ . 
[revised manuscript text omitted]

**List of changes**